# The organellar genomes of *Silvetia siliquosa* (Fucales, Phaeophyceae) and comparative analyses of the brown algae

Yanshuo Liang[1,2,3], Han-Gil Choi[4], Shuangshuang Zhang[1,2,3], Zi-Min Hu[1,2]*, Delin Duan [1,2]*

**1** CAS and Shandong Province Key Laboratory of Experimental Marine Biology, Center for Ocean Mega-Science, Institute of Oceanology, Chinese Academy of Sciences, Qingdao, China, **2** Laboratory for Marine Biology and Biotechnology, Qingdao National Laboratory for Marine Science and Technology, Qingdao, China, **3** University of Chinese Academy of Sciences, Beijing, China, **4** Faculty of Biological Science and Institute for Environmental Science, Wonkwang University, Iksan, Korea

* dlduan@qdio.ac.cn (DD); huzimin9712@163.com (ZMH)

## Abstract

The brown alga *Silvetia siliquosa* (Tseng et Chang) Serrão, Cho, Boo & Brawly is endemic to the Yellow-Bohai Sea and southwestern Korea. It is increasingly endangered due to habitat loss and excessive collection. Here, we sequenced the mitochondrial (mt) and chloroplast (cp) genomes of *S. siliquosa*. *De novo* assembly showed that the mt-genome was 36,036 bp in length, including 38 protein-coding genes (PCGs), 26 tRNAs, and 3 rRNAs, and the cp-genome was 124,991 bp in length, containing 139 PCGs, 28 tRNAs, and 6 rRNAs. Gene composition, gene number, and gene order of the mt-genome and cp-genome were very similar to those of other species in Fucales. Phylogenetic analysis revealed a close genetic relationship between *S. siliquosa* and *F. vesiculosus*, which diverged approximately 8 Mya (5.7–11.0 Mya), corresponding to the Late Miocene (5.3–11.6 Ma). The synonymous substitution rate of mitochondrial genes of phaeophycean species was 1.4 times higher than that of chloroplast genes, but the cp-genomes were more structurally variable than the mt-genomes, with numerous gene losses and rearrangements among the different orders in Phaeophyceae. This study reports the mt- and cp-genomes of the endangered *S. siliquosa* and improves our understanding of its phylogenetic position in Phaeophyceae and of organellar genomic evolution in brown algae.

## Introduction

*Silvetia siliquosa* (Tseng et Chang) Serrão, Cho, Boo & Brawly, a member of the Fucaceae, is an ecologically and commercially important brown alga that occurs in the middle and low intertidal zones. Historically, it has had a wide distribution in the Yellow-Bohai Sea and the southwest coast of Korea [1–4]. However, the natural biomass and distribution range of *S. siliquosa* in East Asia have declined dramatically since the 1990s due to habitat fragmentation and anthropogenic influences [5, 6]. *S. siliquosa* is now listed as an endangered species with a high

**Data Availability Statement:** The datasets analysed during the current study are available in the National Center for Biotechnology Information, and the GenBank accession numbers of the mt-

genome and cp-genome of S. siliquosa are MW485980 and MW485976, respectively. The raw reads of Silvetia siliquosa organelle genomes have been deposited in the NCBI Sequence Read Archive under the BioProject number PRJNA824893, Sequence Read Archive accession number of mitochondrial and chloroplast genomes are SAMN27488512 and SAMN27488513, respectively.

**Funding:** This research was supported by the Strategic Priority Research Program of Chinese Academy of Sciences (XDB42030203, XDB42040106, XDA19060102) (received by DD) and the National Natural Science Foundation of China (Nos. 31971395, 41761144057) (received by Z-MH).

**Competing interests:** The authors have declared that no competing interests exist.

extinction risk in the Yellow-Bohai Sea [7]. Hence, there is an urgent need to restore and conserve its natural populations. For endangered algal species like *S. siliquosa* that have experienced human interference, genomic data will play a fundamental role in effectively preserving their resources and deciphering the factors that endanger them [8]. Limited genomic information, including information on organellar genomes, hampers the conservation of threatened species and the genome-scale evolutionary study of brown algae.

Complete organelle genome data can provide important reference for the phylogenetic construction of brown algae [9, 10]. Compared to the nuclear genome, the relatively simple and conserved structural composition of organellar genomes make them ideal molecular tools for understanding genome evolution across the tree of brown algae [11–13]. Furthermore, the substitution rates of chloroplast genes are generally lower than those of mitochondrial genes [14, 15], and chloroplast genes are therefore more effective for resolving the brown algal phylogeny. Organellar structural variation provides key insights that enhance our understanding of lineage diversification [16]. For example, designing molecular markers based on polymorphism can be used for species identification [10, 17]. Additional organelle genomes from novel taxa will not only provides data support for analyzing the structural variation of organelle genomes, but also advance our understanding of the evolution and diversity of brown algae.

In this study, we sequenced the complete mitochondrial genome (mt-genome) and chloroplast genome (cp-genome) of *S. siliquosa* in order to understand its organellar genomic architecture and preserve its genome resources. We explored the evolutionary status of *S. siliquosa* in Phaeophyceae at the mt-genome level and the divergence time of typical brown algae. We also compared the organellar genomes of *S. siliquosa* and other typical brown algae to determine how structures and substitution rates varied across organelles and lineages.

## Materials and methods

### Algal material and DNA extraction

*Silvetia siliquosa* was collected from the rocky shore on Jindo Island, Korea (34˚40'N, 126˚28'E) in 2018. *S. siliquosa* is not listed on any Asian official threatened species list due to weak legislation and less research on endangered seaweeds. No special permits were required for this study and the sample was collected by researchers from Wonkwang University of Korea. To avoid damage to algae, the tip of apical vegetative tissue (3–5 cm) was excised and stored in silica gel. The total genomic DNA was extracted using the FastPure Plant DNA Isolation Mini Kit (Vazyme Biotech Co., Ltd., Nanjing, China) according to the manufacturer's instructions. The extracted DNA was subsequently purified based on quality control protocols.

### Illumina sequencing, genome assembly and annotation

After DNA purification, 1 μg of DNA was used to construct paired-end libraries with insert sizes of 450 bp following Illumina's standard genomic DNA library preparation procedure. The quality-checked Illumina paired-end libraries were sequenced on the Illumina HiSeq 4000 platform (Biozeron, Shanghai, China). The raw paired-end reads were trimmed and quality-filtered using Trimmomatic-0.39 [18] with parameters SLIDINGWINDOW: 4:15 MINLEN: 75. Clean data obtained after quality control were used for further analysis. We used SOAPde-novo v2.04 [19] to construct *de novo* assemblies, and contigs were Blasted against the reference organellar genomes of *Fucus vesiculosus* (mt-genome: NC_007683; cp-genome: FM957154). Aligned contigs with high similarity (≥ 80%) were ordered based on the reference genomes. GapCloser v1.12 [19] was subsequently used to fill in the remaining local inner gaps. Finally, an mt-genome with one 36,036 bp scaffold and a cp-genome with one 124,991 bp scaffold were obtained.

Protein-coding genes (PCGs) and open reading frames (ORFs) were annotated using the online Dual Organellar GenoMe Annotator tool (DOGMA) with default parameters [20]. The transfer RNA (tRNA) genes were identified by reconstructing their cloverleaf structures using tRNAscan-SE v1.23 with default parameters [21], and ribosomal RNA (rRNA) genes were determined using RNAmmer v1.2 [22]. The circular mitochondrial and chloroplast genomes map were drawn using OGDRAW v1.3.1 [23]. The mt-genome and cp-genome of *S. siliquosa* were deposited in GenBank under accession numbers MW485976 and MW485980, respectively.

## Boundary regions and synteny analysis

To identify possible structural rearrangements in organellar genomes of the Phaeophyceae, we used Mauve [24] to conduct co-linear analysis with the following settings: progressive Mauve alignment algorithm, the organellar genomes of *S. sililiquosa* as the reference sequences, and automatic calculation of full alignment and minimum locally collinear block (LCB) score. To detect variations in the LSC/IR/SSC boundaries of the chloroplast genomes in Phaeophyceae, we compared and visualized the exact IR border positions and their adjacent genes using the online tool IRscope [25].

## Phylogenetic analysis and divergence timing

Phylogenetic relationships within the Phaeophyceae were analyzed using the concatenated sequence datasets of 35 shared mitochondrial PCGs (*rps2–4*, *rps7*, *rps8*, *rps10–14*, *rps19*; *rpl2*, *rpl5*, *rpl6*, *rpl14*, *rpl16*, *rpl31*; *nad1–7*, *nad9*, *nad11*; *cob*; *cox1–3*; *atp6*, *atp8*, *atp9*; and *tatC*) from 19 brown algae (see S1 Table). The nucleotide sequences of each gene were aligned using default setting of ClustalW 2.0 [26] and then concatenated for tree construction. *Heterosigma akashiwo* (Raphidophyceae, GenBank number: NC_016738) was selected as an outgroup. Maximum likelihood (ML) and Bayesian inference (BI) trees were reconstructed using PhyML v.3.1 [27] and MrBayes v.3.2 [28], respectively. Modeltest v3.7 [29] was used to determine the best-fit substitution model for the concatenated dataset (GTR+G+I, I = 0.1682, G = 0.5123) under the Akaike information criterion (AIC). The ML tree was constructed based on Sub-tree-Pruning-Regrafting (SPR) with heuristic analysis of $10^3$ bootstrap replicates. For BI analysis, the Markov Chain Monte Carlo (MCMC) process was run for $2\times10^6$ generations using four chains with a tree sampling frequency of every 200 generations, discarding the first 10% as burn-in and calculating the posterior consensus tree.

We concatenated five mitochondrial genes (*cox1*, *cox3*, *nad1*, *nad4*, and *atp9*) and three chloroplast genes (*rbcL*, *psbA*, and *atpB*) from 15 brown algae for molecular dating. These genes were highly conserved and slow-evolving. After alignment, the concatenated sequences were divided into three partitions corresponding to the 1st, 2nd, and 3rd codon sites. ML trees were reconstructed using PhyML v.3.1 based on the best scoring alternative model of GTR+G +I with 100 bootstrap replicates. Divergence times were estimated by the approximate likelihood calculation method implemented in MCMCTree of PAML v4.8 [30, 31]. Two fossil calibrations were incorporated based on previous studies (S2 Table). The prior parameters of rgene_gamma was calculated with estimates of the overall substitution rate on the ML tree obtained by BASEML in PAML. The gradient and Hessian of the branch lengths were estimated by BASEML using the GTR+G substitution model at the maximum likelihood estimates [30]. The independent rate model (clock = 2) for the molecular clock and the GTR+G model for nucleotide substitutions were set in the mcmctree.ctl control file, with the following parameter settings: substitution rate per time unit = 0.080406; rgene_gamma = 1 12.5; sigma2_-gamma = 1 4.5. To determine whether convergence had been achieved, two independent MCMC chains were run with $5\times10^6$ steps after discarding $10^4$ generations as burn-in.

**Table 1. General features of mitochondrial genomes in Phaeophyceae.**

| Genome Features | *Silvetia siliquosa* | *Fucus vesiculosus* | *Sargassum thunbergii* | *Sargassum horneri* | *Desmarestia viridis* | *Saccharina japonica* | *Undaria pinnatifida* | *Ectocarpus siliculosus* | *Dictyota dichotoma* |
|---|---|---|---|---|---|---|---|---|---|
| Genome Size / GC Content (%) | 36,036/33.75 | 36,392/34.45 | 34,748/36.62 | 34,606/36.16 | 39,049/36.60 | 37,657/35.30 | 37,402/32.53 | 37,187/33.51 | 31,617/36.52 |
| Gene number rRNA/ tRNA/CDS/ Total | 3/26/38/67 | 3/26/38/67 | 3/25/37/65 | 3/25/37/65 | 3/26/39/68 | 3/25/38/66 | 3/25/38/66 | 3/25/40/68 | 3/25/38/67 |
| Total Gene Length | 27,879 | 28,212 | 27,096 | 27,060 | 30,570 | 29,007 | 29,067 | 28,764 | 24,513 |
| Average Gene Length | 734 | 742 | 732 | 731 | 784 | 763 | 765 | 719 | 645 |
| Gene's GC Content | 32.5 | 33.3 | 35.5 | 35.1 | 35.6 | 34.2 | 31.1 | 32.3 | 35.5 |
| % of Genome (Genes) | 77.36 | 77.52 | 77.98 | 78.19 | 78.29 | 77.03 | 77.72 | 77.35 | 77.53 |
| Intergenic region length | 8,157 | 8,180 | 7652 | 7546 | 8479 | 8650 | 8335 | 8423 | 7104 |
| % of Genome (Intergenic) | 22.64 | 22.48 | 22.02 | 21.81 | 21.71 | 22.97 | 22.28 | 22.65 | 22.47 |
| Spacer content (%) | 5.69 | 5.61 | 4.12 | 4.29 | 6.06 | 6.49 | 5.83 | 6.34 | 3.21 |
| Spacer size (bp) | 0–209 | 0–422 | 0–166 | 0–172 | 0–385 | 0–361 | 0–354 | 0–356 | 0–74 |
| Pairs of overlapping genes | 10 | 10 | 14 | 12 | 13 | 13 | 15 | 14 | 12 |
| Overlap size (bp) | 1–66 | 1–66 | 1–60 | 1–66 | 1–60 | 1–16 | 1–60 | 1–59 | 1–30 |
| GenBank accession | MW485980 | NC_007683 | NC_026700 | NC_024613 | NC_007684 | NC_013476 | NC_023354 | FP885846 | NC_007685 |

## Substitution rate estimation

To investigate the variation in nucleotide substitution rates of mt- and cp-genomes in the Phaeophyceae, we retrieved 35 mitochondrial PCGs and 129 chloroplast PCGs to measure the ratio of non-synonymous (dN) and synonymous substitutions (dS). We selected the brown algae listed in Tables 1 and 2 for this analysis. We performed codon alignment for each PCG using MEGA and identified conserved blocks using Gblocks v0.91b with default parameters [32]. The alignment sequence was transformed into pml format using DAMBE5 [33]. We estimated dN, dS, and dN/dS ratio using the Codeml program in PAML v4.8 [31] with the following options: runmode = −2 and CodonFreq = 2. Genes with synonymous substitution values greater than 5 were discarded from further analysis. The dN/dS values were averaged for all pairwise comparisons of each gene. The significance of differences between mean values was determined by independent-samples *t*-test with a 95% confidence interval using SPSS software.

## Results and discussion

### The mitochondrial genome of *S. siliquosa*

The circular mt-genome of *S. siliquosa* is 36,036 bp in length (Fig 1), longer than those of *Dictyota dichotoma*, *Sargassum thunbergii*, and *Sargassum horneri* but shorter than those of the Fucophycidae (Table 1). Its overall GC content of 33.75% is comparable to those of other phaeophycean species (i.e., 32.53–36.60%, Table 1). The mt-genome of *S. siliquosa* is gene dense, and the length of coding genes accounts for 94.31% of the total mt-genome, and non-coding regions accounts for only 5.69%, well within the range of Phaeophycean species (3.21–6.49%, Table 1). An overlap of base A is present between *rpl6* and *rps2* in the mt-genome of *S. siliquosa*, and the overlapping regions are exceedingly conserved in eight mt-genomes of the

**Table 2. General features of chloroplast genomes in Phaeophyceae.**

| Genome Features | *Silvetia siliquosa* | *Fucus vesiculosus* | *Sargassum horneri* | *Sargassum thunbergii* | *Saccharina japonica* | *Costaria costata* | *Undaria pinnatifida* | *Ectocarpus siliculosus* | *Dictyopteris divaricata* |
|---|---|---|---|---|---|---|---|---|---|
| Genome Size / GC content (%) | 124,991/ 28.84 | 124,986/28.9 | 124,068/ 30.61 | 124,592/30.40 | 130,584/31.05 | 129,947/ 30.87 | 130,383/30.61 | 139,954/30.67 | 126,099/31.19 |
| LSC size (bp) / GC content (%) | 74,247/27.32 | 74,287/27.42 | 73,311/29.20 | 73,668/29.00 | 77,379/29.79 | 76,507/29.72 | 76,598/29.53 | 80,011/29.42 | 72,648/30.14 |
| SSC size (bp) / GC content (%) | 40,222/27.93 | 40,215/27.98 | 40,139/29.74 | 40382/29.48 | 43,175/30.04 | 42,622/29.67 | 42,977/29.26 | 42,711/29.77 | 41,455/30.26 |
| IR size (bp) / GC content (%) | 10,522/43.07 | 10,484/43.32 | 10,618/43.67 | 10,542/43.77 | 10,030/44.29 | 10,818/43.73 | 10,808/43.71 | 17,232/38.70 | 11996/40.75 |
| Gene number rRNA/ tRNA/CDS/ Total | 6/28/140/174 | 6/28/140/ 174 | 6/28/140/174 | 6/28/140/174 | 6/29/141/176 | 6/27/141/174 | 6/28/141/175 | 6/31/148/185 | 6/28/138/174 |
| intron | 1 | 1 | 2 | 1 | 1 | 1 | 1 | 0 | 1 |
| Total Gene Length | 96,216 | 96,183 | 95,907 | 95,805 | 97,971 | 97,935 | 97,920 | 101,154 | 97,641 |
| Average Gene Length | 687 | 687 | 685 | 684 | 695 | 695 | 694 | 683 | 708 |
| Gene's GC Content (%) | 29.34 | 29.42 | 30.90 | 30.83 | 31.69 | 31.51 | 31.36 | 31.56 | 31.59 |
| % of Genome (Genes) | 76.98 | 76.96 | 77.30 | 76.89 | 75.03 | 75.37 | 75.10 | 72.28 | 77.43 |
| Intergenic region length | 28,775 | 28,803 | 28,161 | 28,787 | 32,613 | 32,012 | 32,463 | 38,800 | 28,458 |
| % of Genome (Intergenic) | 23.02 | 23.04 | 22.70 | 23.11 | 24.97 | 24.63 | 24.90 | 27.72 | 22.57 |
| Genes duplicated in IR | 5 | 5 | 5 | 5 | 5 | 5 | 5 | 11 | 6 |
| GenBank accession | MW485976 | FM957154 | NC_029856 | NC_029134 | JQ405663 | NC_028502 | NC_028503 | FP102296 | KY433579 |

Phaeophyceae [11–13]. In addition, there are two more highly conserved overlapping regions in the Fucophycidae species: ATGA, which overlaps with *rps8* and *rpl6*, and ATGCTCTTAA, which overlaps with *cox2* and *nad4*. However, the *rps8-rpl6* overlap in *D. dichotoma* is 7 bp in length (GTGGTAA), and there are no overlaps between *cox2* and *nad4* in *D. dichotoma*.

The mt-genome of *S. siliquosa* contains 38 PCGs (including 3 conserved ORFs), 3 rRNAs (*rnl*, *rns*, and *rrn5*), and 26 tRNAs. None of the genes contain introns. There are 64 conserved homologous genes (3 rRNAs, 24 tRNAs, and 37 PCGs, including 2 ORFs) that are also observed in nine brown algal mt-genomes, underscoring the highly conserved gene content of these genomes. *TrnM-2* is located between *trnQ* and *ORF39* in the order Fucales and *E. siliculosus*, but *trnI* is located here in *D. viridis*, *S. japonica*, *U. pinnatifida* and *D. dichotoma*. In addition, *trnL-3* is only found between *trnA* and *rps10* in *S. siliquosa*, and it has been replaced by *trnY-2* in *F. vesiculosus* and *D. viridis*. *S. siliquosa* and *F. vesiculosus* share two conserved ORFs, despite their different sizes (*ORF39* and *ORF43*, *ORF331* and *ORF379*).

All PCGs encoded by the *S. siliquosa* mt-genome have a methionine (ATG) as the start codon, with the exception of *ORF331*, which has a TTG. This phenomenon has been reported in other brown algal mt-genomes. For example, *ORF221* in *D. viridis* and *ORF37* in *D. dichotoma* use TTG as the start codon, whereas *ORF379* in *F. vesiculosus* uses GTG as the start codon [11]. Three stop codons are used, with a preference of 84.21% for TAA (10.53% for TAG and 5.6% for TGA). This is similar to other reported brown algal mt-genomes, although the proportions are slightly different [11–13, 34].

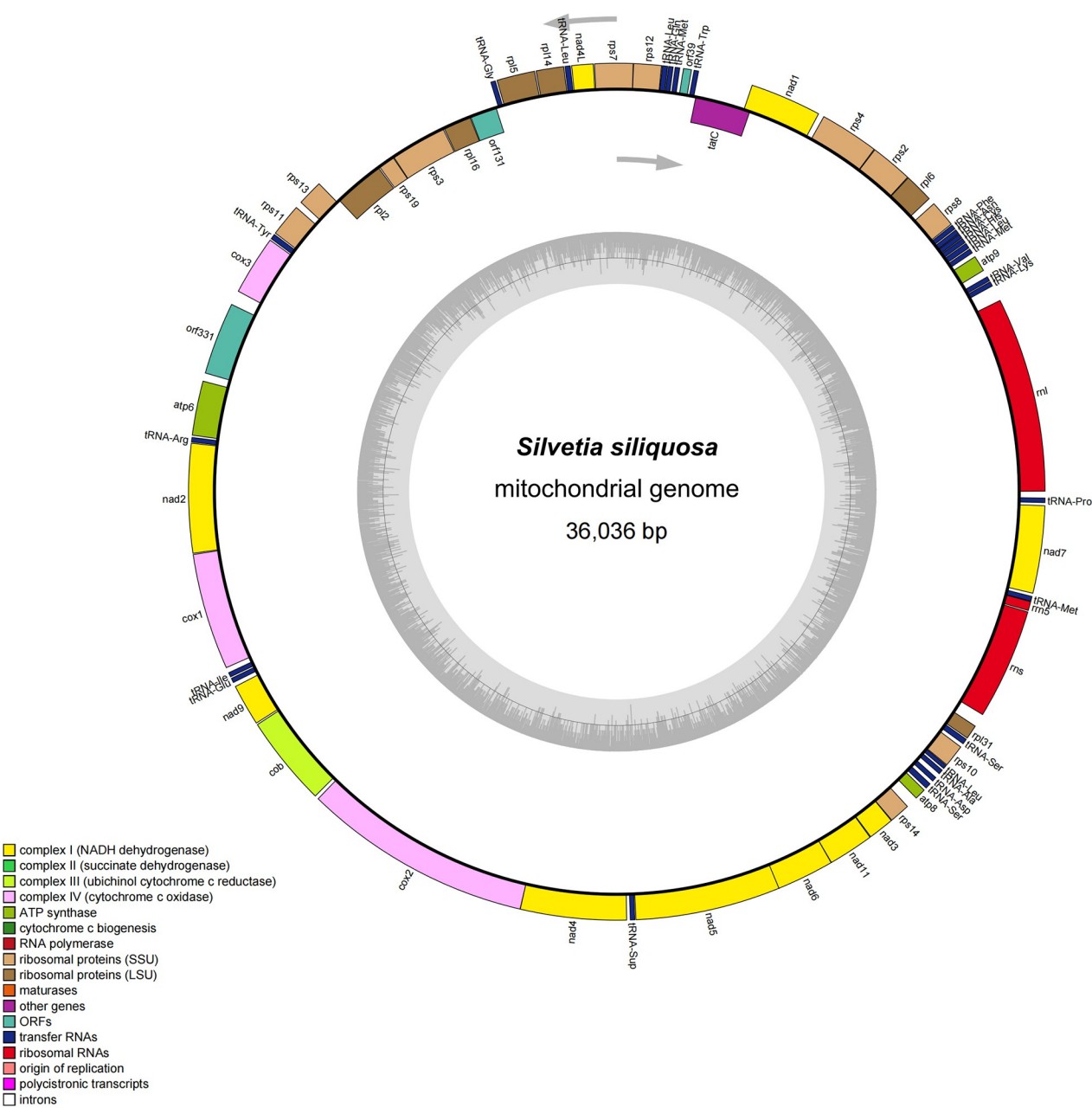

**Fig 1. The mitochondrial genome of *S. siliquosa*.** Annotated genes are colored according to the functional categories. Genes on the outside are transcribed in the clockwise direction, whereas genes on the inside are transcribed in the counterclockwise direction.

## The chloroplast genome of *S. siliquosa*

The cp-genome of *S. siliquosa* is a circular molecule of 124,991 bp (Fig 2). It is the largest cp-genome in the Fucales (124,068–124,986 bp) but smaller than those of *E. siliculosus* (139,954 bp), *D. divaricata* (126,099 bp), and species in the Laminariales (129,947–130,584 bp, Table 2). Its GC content (28.84%) is lower than that of other brown algal cp-genomes, which range from 28.94% (*F. vesiculosus*) to 31.19% (*D. divaricata*) (Table 2). The cp-genome of *S. siliquosa* displays a canonical quadripartite structure with two large inverted repeats of 5,261 bp divided

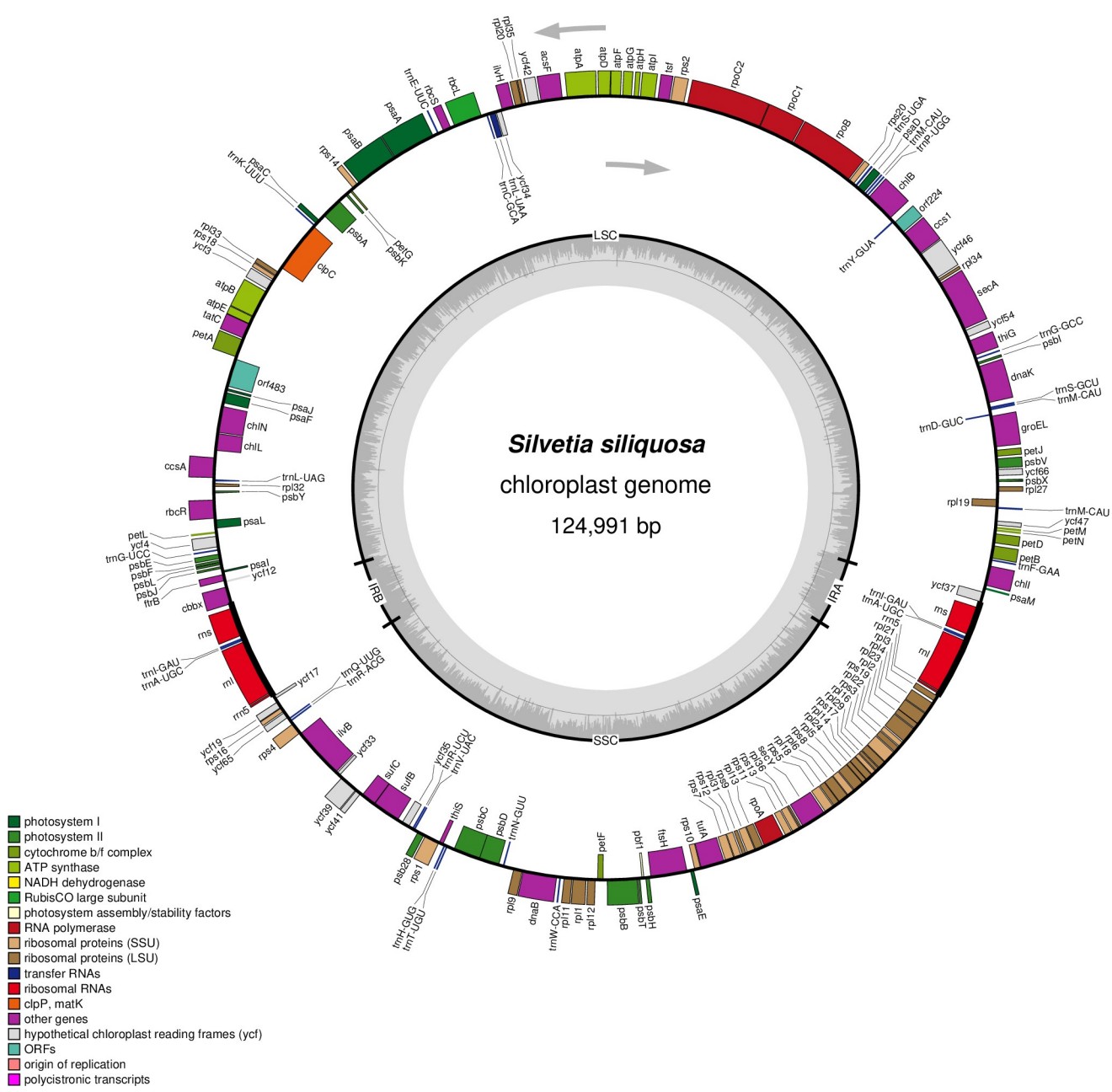

**Fig 2. The chloroplast genome of S. siliquosa.** Annotated genes are colored according to the functional categories. Genes on the outside are transcribed in the clockwise direction, whereas genes on the inside are transcribed in the counterclockwise direction.

by a short single copy region (SSC, 40,222 bp) and a long single copy region (LSC, 74,247 bp) (Table 2). The GC content of the IR regions (43.07%) is higher than that of the LSC (27.32%) and the SSC (27.93%). Protein-coding sequences constitute 76.98% of the cp-genome of *S. siliquosa*, similar to other cp-genomes in the Phaeophyceae (72.28–77.43%). The IRs of *S. siliquosa* are composed of the core *rrn5-rnl-trnA-trnI-rns* gene cluster, which is similar to those in Fucales and Laminariales [35–38] but different from those of *E. siliculosus* and *D. divaricata*, which have longer IRs (11,996–17,232 bp) and contain 11 and 6 gene loci, respectively [9, 39].

The cp-genome of *S. siliquosa* contains 174 genes, including 140 PCGs, 28 tRNAs, and 6 rRNAs (Table 2). Only one intron is found in *trnL-2*, and this intron is also existed in the homologous genes of Phaeophycean species, but absent in *E. siliculosus* [35, 36, 40].

All cp-genomes in Phaeophyceae share a core set of 136 genes, underscoring the high structural conservation of cp-genomes in the brown algae (S1 Fig). However, the four species (*S. siliquosa*, *F. vesiculosus*, *S. horneri*, and *S. thunbergii*) in Fucales are missing the *syfB* gene that is present in *D. divaricata*, Laminariales, and Ectocarpales. We speculate that this gene may have been lost in a common ancestor of the order Fucales, although more taxonomic groups must be added to confirm this possibility. The *syfB* gene encodes the *β* subunit of phenylalanyl-tRNA synthetase [17], and its loss may affect the synthesis of *trnF* encoded in the cp-genomes [40]. Moreover, three PCGs (*Escp36 = Escp99*, *Escp117*, and *Escp161*) are found only in *E. siliculosus* but absent in other species. The *rpl32* and *rbcR* genes have been lost in the cp-genome of *D. divaricata* but are present in Fucales, Laminariales, and *E. siliculosus*. The absence of these genes may be due to gene transfer to the nucleus or gene loss [39]. All the PCGs begin with an ATG codon with the exception of *psbF* in *S. siliquosa*, which begins with a GTG; 116 PCGs are terminated by a TAA stop codon, 16 by a TAG, and 8 by a TGA.

## Phylogenetic assessment and molecular dating of brown algae

The phylogenetic dataset included 35 PCGs from the mt-genomes of 19 phaeophycean species, and the total length of the concatenated sequence alignment was 23,604 bp. *H. akashiwo* was used as the outgroup. Congruent topologies were obtained from maximum likelihood and Bayesian inference on the complete data set, and all branches exhibited a high support rate (S2 Fig). Phylogenetic trees showed that 19 species of brown algae fit well into five established clades: Fucales, Laminariales, Ectocarpales, Desmarestiales, and Dictyotales. *S. siliquosa* and the two species of the genus *Fucus* (*F. vesiculosus* and *F. distichus*) formed sister groups with high bootstrap support values. The Fucales species diverged later in the Phaeophyceae, and their divergence was significantly later than those of Laminariales, Ectocarpales, Desmarestiales, and Dictyotales. The reconstructed phylogenetic tree supported Laminariales and Ectocarpales as sister monophyletic groups (S2 Fig). However, a previous phylogenetic tree of these brown alga based on three rRNA genes (*rnl*, *rns*, and *rrn5*) indicated that Laminariales formed a monophyletic group with Desmarestiales, and this group was sister to the Ectocarpales group [12]. These topological differences may be due to the different evolutionary rates of rRNA regions and protein coding gene regions. Here, Dictyotales diverged first in the Phaeophyceae and had a sister relationship with other Phaeophyceae species. This result is consistent with previous studies [11, 12].

Due to the incompleteness of the mitochondrial and chloroplast gene data sets, we reconstructed a phylogenetic tree of 15 brown algae species using a concatenated sequence of five mitochondrial genes (*cox1*, *cox3*, *nad1*, *nad4*, and *atp9*) and three chloroplast genes (*rbcL*, *psbA*, and *atpB*). Fossils of a *Padina*-like morphology have been found in the Early Cretaceous (145.5–99.6 Mya) clay shales [41, 42], and we therefore defined a lower boundary at 99.6 Mya for the stem node of *Padina boryana*. In addition, a few species of Cystoseiraceae have been found in the Monterey deposit (17–13 Mya) [43], and the crown node of the Fucales was therefore given a minimum age of 13 Mya [44]. Two run results based on the maximum likelihood method were very similar, and we concluded that they had achieved convergence [30]. Time-calibrated molecular clock analyses suggested that *S. siliquosa* and *F. vesiculosus* began to diverge approximately 8 million years ago (5.7–11.0 Mya based on 95% highest posterior densities, HPD) in the Late Miocene (5.3–11.6 Mya) (Fig 3). This was similar to the results of Silberfeld et al. (2010), although the species they used were *F. vesiculosus* and *Pelvetia*

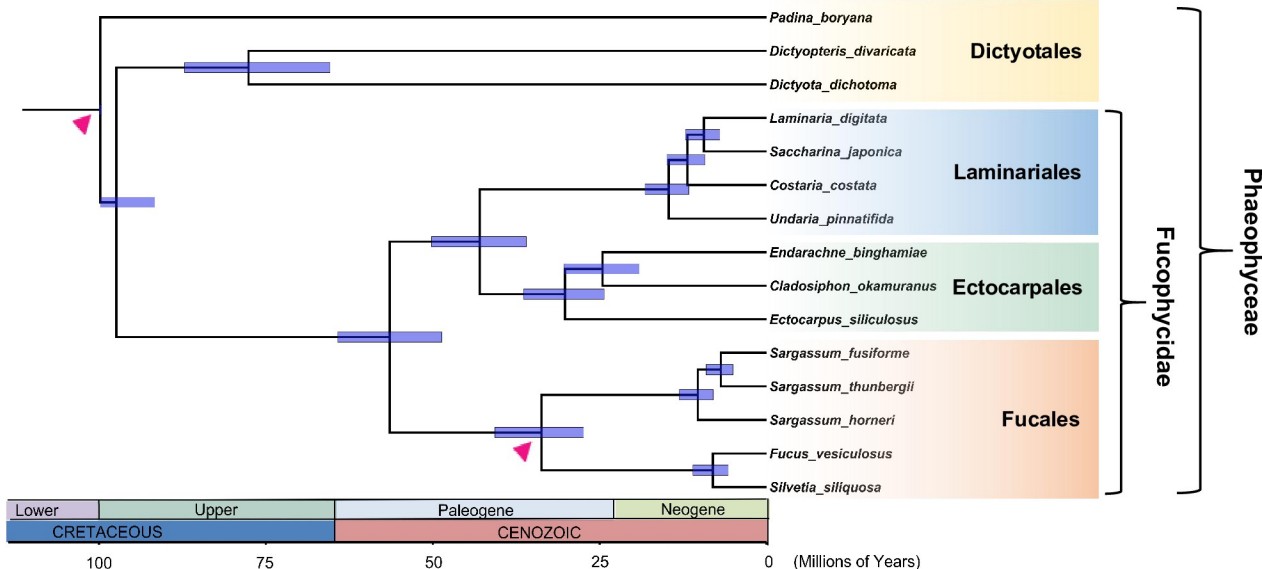

**Fig 3. Posterior estimates of divergence time of 15 taxa on the phylogenetic tree.** Blue bars depict the 95% highest posterior density (HPD) and the values at the nodes represent posterior mean ages. Estimations were performed with MCMCTree based on the independent rate model using two fossil calibrations on nodes indicated by arrows.

*canaliculata*, and *S. siliquosa* belonged to the genus *Pelvetia* before 1999 [45]. According to the time-calibrated clock, four brown algal orders diversified from the Upper Cretaceous to the Paleocene, and the diversifications of the Fucales, the Laminariales–Ectocarpales clade, and the Dictyotales began approximately 33.7 Mya, 56.4 Mya and 97.4 Mya, respectively. Yip et al. (2020) also selected Sargassaceae and Fucaceae species and estimated divergence times; they obtained one 95% HPD interval between the two families at 16.4–39.4 Mya [46], which over-laps with the average age of diversification (33.7 Mya) inferred in this study. The previous esti-mate for the average time of divergence between Laminariales and Ectocarpales was 98.0 Mya [42], significantly earlier than our estimate (36.0–50.2 Mya, 95% HPD). This difference may reflect the addition of a more distinct outgroup in the previous study, which may have caused this node to be pushed forward [44]. Due to limited fossil data for Phaeophyta [42], it is not surprising that the uncertainty of the divergence and diversification dates of brown algae spans several million years [47].

## Substitution rate estimation

We estimated the synonymous and non-synonymous substitution rates based on the ML method implemented in PAML. This is the most accurate method currently available to mea-sure substitution rates [31, 48], and by measuring the synonymous substitution rate (dS) in mt-genomes and cp-genomes of closely related species, we can obtain the relative mutation rate between them [49]. The average dS values in Phaeophyceae varied from 0.845 to 4.715 for mitochondrial genes and from 0.435 to 3.151 for chloroplast genes (Fig 4C; S3 Table). Mito-chondrial and chloroplast protein-coding genes differed significantly in synonymous substitu-tion rate based on an independent-sample $t$-test (p<0.001), and the mitochondrial mutation rate was 1.4 times that of the chloroplast. The average nonsynonymous substitution rate (dN) was significantly higher in mt-genomes than in cp-genomes (p<0.05). Specifically, the values for mt-genome genes were 1.3-fold higher than those for cp-genome genes (Fig 4B; S4 Table). The non-synonymous/synonymous rate ratio (dN/dS) is an important indicator used to infer

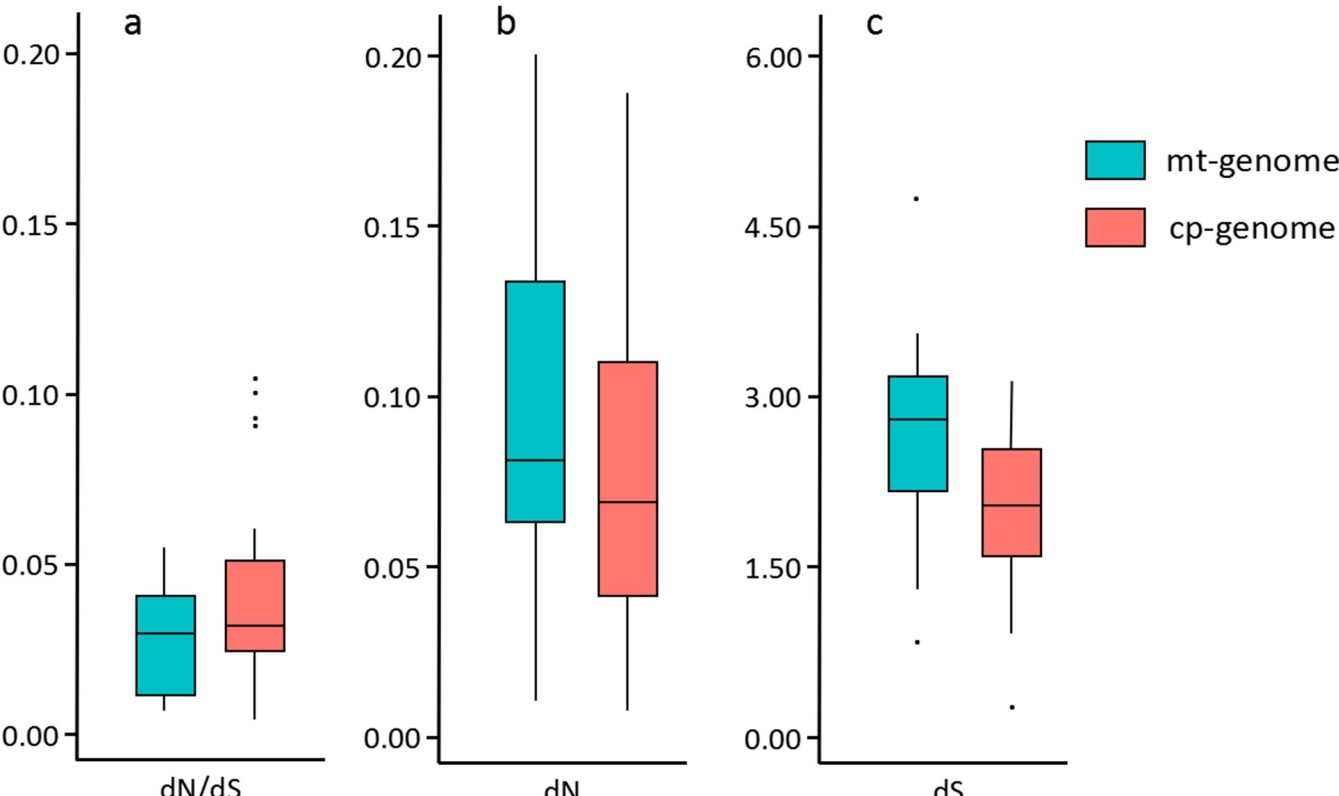

**Fig 4. Boxplots showing synonymous substitutions (dS), nonsynonymous substitutions (dN), and dN/dS ratios in mt- and cp-genomes in Phaeophyceae.** The box represents the values between the quartiles. Outliers are shown as black points, and the black lines inside the box represent the median values.

the selection pressure at the protein level [50]. The dN/dS ratios were similar and less than 1 in the genomes of both organelles in phaeophycean species, indicating that protein-coding genes in the mt- and cp-genomes have been subjected to stronger purifying selection (Fig 4A; S3 and S4 Tables). The higher substitution rate observed in mitochondrial protein-coding genes is the result of the high mutation rate caused by the presence of oxygen free radicals in mitochondria [51]. Previous studies have also found that the substitution rates of mitochondrial genes in green algae *Volvulina compacta* and the red algal genus *Porphyra* are greater than those of chloroplast genes [52, 53]. By contrast, the opposite result is observed in most seed plants, in which the mitochondrial substitution rate is estimated to be lower than that of the chloroplast [51]. Although the consequences of markedly different substitution rates between the two genomes are not fully understood, they are likely to reflect the evolutionary history of organelle genomes among different lineages.

## IR contraction and expansion

When we compared the IRb/LSC junctions (JLB) of cp-genomes in the Fucales, we did not find major variations in the IR regions of *S. siliquosa*, *S. horneri*, and *S. thunbergii*. Their IRb boundaries extended to the *cbbx* gene (Fig 5), and the extension varied from 59 bp (*S. horneri*) to 188 bp (*S. thunbergii*). However, the *cbbx* gene of *F. vesiculosus* is located in the LSC region, 141 bp away from the JLB border, and *F. vesiculosus* showed a significant contraction in the IR region (4,863 bp) among other members of the order Fucales (Fig 5). The IRb/SSC junctions (JSB) of IRb were located mainly between *rrn5* (plus strand) and *ycf19*, but the IRb boundary

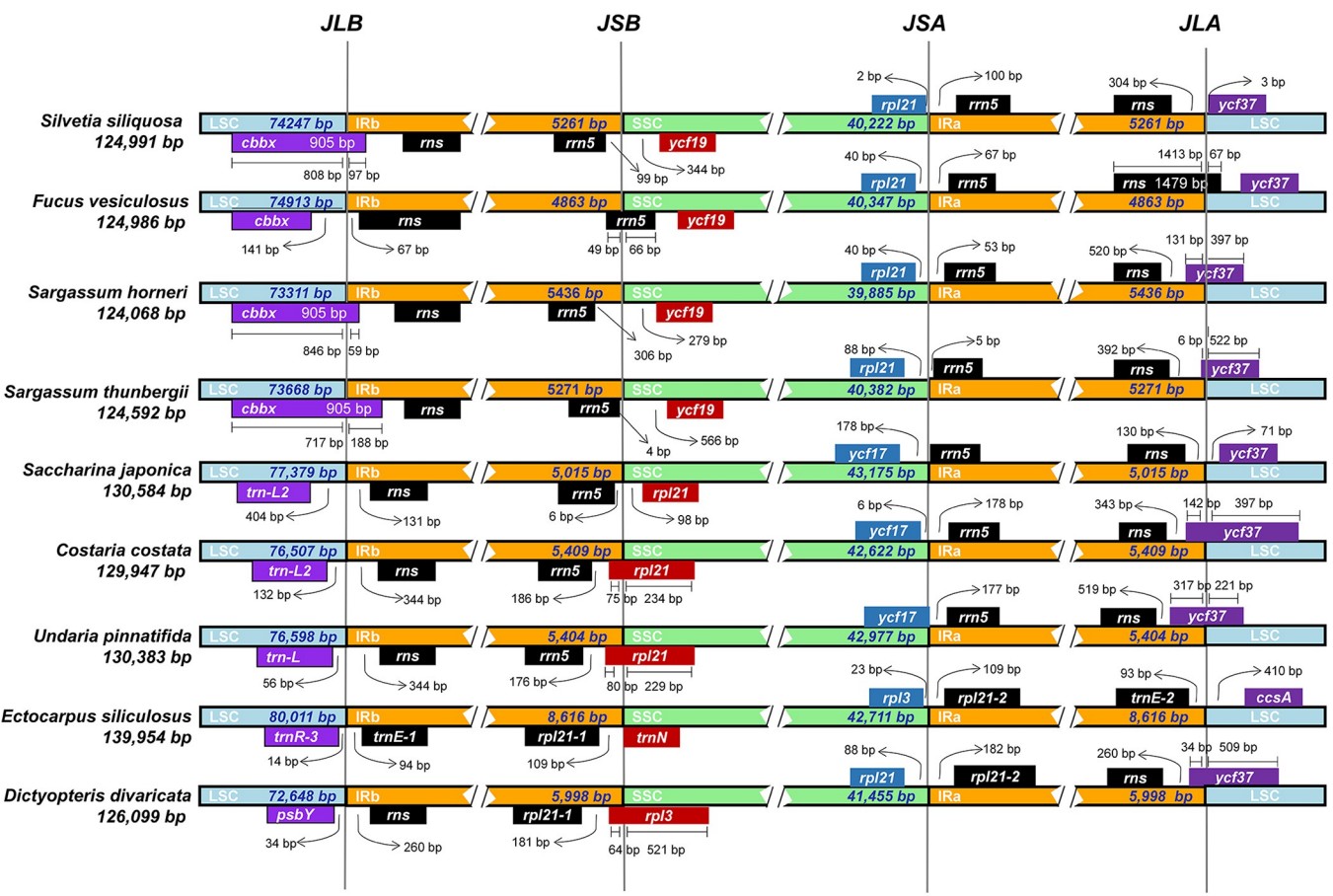

**Fig 5. Comparison of the borders of LSC, SSC and IR regions among Phaeophyceae chloroplast genomes.**

of *F. vesiculosus* extended into *rpl21*. The *rpl21-rrn5* (minus strand) sequences are located at the junction of the SSC/IRa regions (JSA) in the four Fucales species. However, *S. horneri* and *S. thunbergii* have a longer IR region with only minor expansions, and their IRa regions extend into the *ycf37* gene (Fig 5).

Through comparison of the IR boundary regions of Fucales, Laminariales, Ectocarpales, and Dictyotales, we found that the IR boundaries in Phaeophyceae vary considerably at the order level (Fig 5). *S. siliquosa* and *S. japonica* are the most similar at the IRa/LSC boundary (JLA), which is located between *rns* and *ycf37* in both species, whereas that of *C. costata*, *U. pinnatifida*, and *D. divaricata* extends into *ycf37*. Interestingly, *E. siliculosus* is quite different from other brown algae, and its JLA conjunction expands into the region between *trnE-2* and *ccsA*. We speculated that variation in the JLA boundary may not be related to the phylogeny of the lineage. Unlike that of Fucales species, the JLB boundary region of the Laminariales is located between *trnL* and *rns*, and the JSA boundary region is located between *ycf17* and *rrn5*. The *rpl21* sequence is found at the JSB boundary in most species of Laminariales, with the exception of *S. japonica*. The contraction and expansion of the IR boundary may be the result of gene conversion and double-strand break recombination repair [54], which is a primary reason for size changes in cp-genomes [39]. A previous report noted that expansions of the IR may be involved in the emergence and diversification of monocot angiosperms [55]. Therefore, we speculated that variation at the cp-genome structure level may play an important role in the divergence of Phaeophyceae species.

## Collinearity analysis of organellar genomes

By analyzing local collinear blocks among the brown algae, we found that mt-genomic architecture was conserved. Only one rearrangement was found in *D. dichotoma* (S3 Fig), and it involved the displacement of *atp8*, *rpl31*, *rps10*, and *atp9*. Although these phaeophycean brown algae represent a variety of morphologically divergent taxa and have a long evolutionary history, most exhibit conserved mt-genome synteny with little variation in gene composition. This is because *D. dichotoma* is an early divergent lineage, and the remaining orders have experienced the brown algal crown radiation (BACR) followed by strong constraints on mitochondrial gene content and genome evolution [42, 56]. This conserved mt-genome structural pattern has also been reported in the Florideophyceae [57].

By contrast, the brown algal cp-genomes demonstrated many rearrangements and inversion events at the order level (Fig 6). Syntenic regions of the four cp-genomes in the Fucales (*S. siliquosa*, *F. vesiculosus*, *S. horneri*, and *S. thunbergii*) showed no rearrangements relative to one another. Similarly, three Laminariales species (*S. japonica*, *C. costata*, and *U. pinnatifida*) showed identical genome architecture. This indicates that no recombination events occurred after the divergence of orders in Phaeophyceae. However, cp-genomes in the Fucales and Laminariales exhibited several rearrangements, and the number of rearrangements in the *E. siliculosus* cp-genome was twice as high as that in other brown algae (Fig 6). Interestingly, structural variations in the cp-genomes of the Laminariales and *E. siliculosus* were not correlated with their phylogenetic relationships (S2 Fig), and the collinearity between species of the Laminariales, Fucales and Dictyotales was higher than between any of these groups and *E. siliculosus*. Recent research has found that variation in the chloroplast architecture of Ectocarpales species may be linked to their reproductive strategy and mode of organellar inheritance [16]. The chloroplast genomes of Ectocarpales species with biparental inheritance show greater structural variation than those of other brown algal lineages, and many brown algae adopt

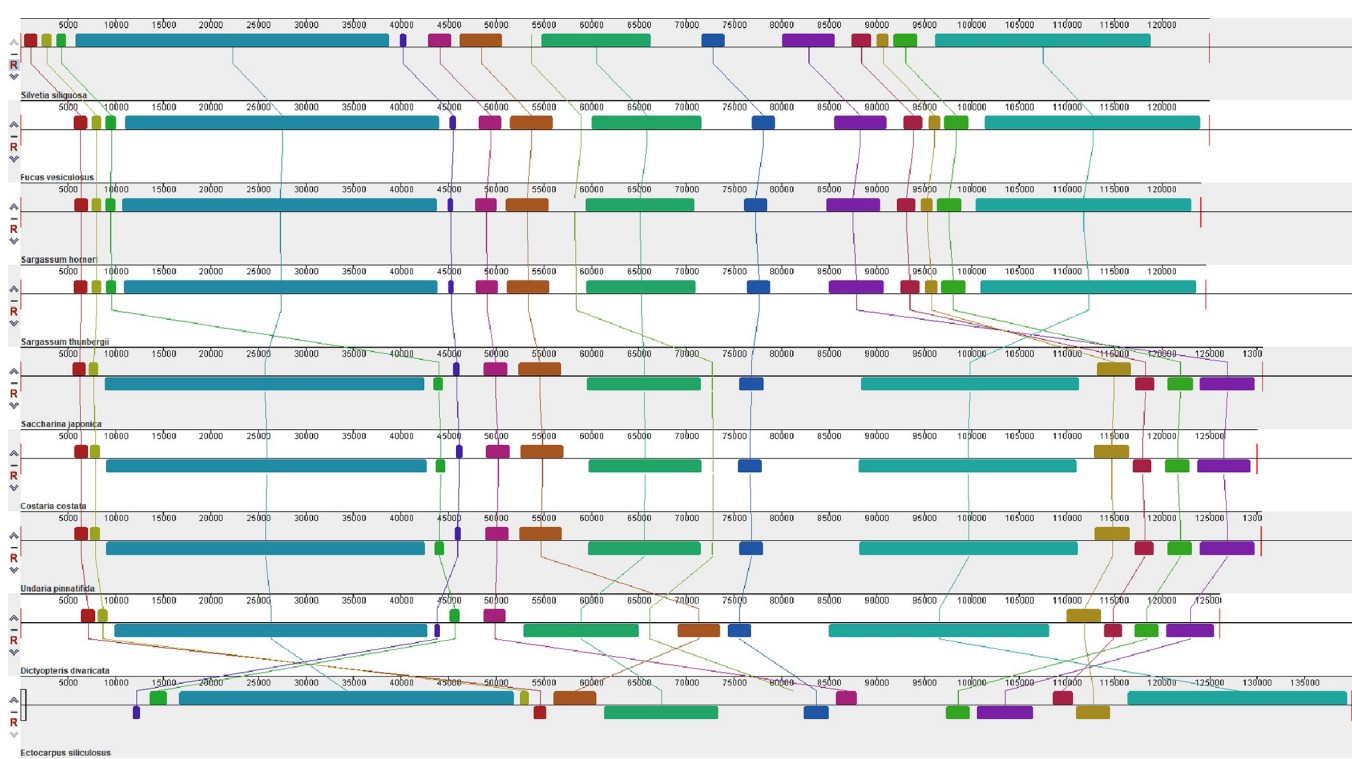

**Fig 6. The collinearity analysis of Phaeophyceae chloroplast genomes.**

maternal inheritance for oogamous reproduction [16]. This genetic pattern and chloroplast structure rearrangement coupling event is supported in another phaeophycean order Sphacelariales [58]. Furthermore, we believe that a large number of structural rearrangements at the order level may play an important role in the process of species divergence. Although we did not verify this possibility, a recent study has shown that rearrangements of two IR-flanking inverted fragments in Taxaceae species were involved in the divergence of this family [59]. However, it will be necessary to obtain more chloroplast genome data to fill in the gaps and fully understand species structural evolution in the Phaeophyceae.

## Conclusions

We sequenced and analyzed the mitochondrial and chloroplast genomes of the threatened species *Silvetia siliquosa* for the first time. The structures of *S. siliquosa* organellar genomes were highly similar to those of *F. vesiculosus*, and we estimated the divergence time between *S. siliquosa* and *F. vesiculosus* for the first time based on fossil correction. We also analyzed the substitution rates and structural variations of mt-genomes and cp-genomes among phaeophycean algae. The results suggested that the synonymous substitution rate of mt-genomes was significantly higher than that of cp-genomes, but a large number of structural variations were detected among cp-genomes at the order level, and these structural changes may be related to species diversification. However, our study did not integrate all brown algae orders, and additional organellar genomes at the ordinal level are needed for further study of their organellar genome evolution.

## Supporting information

**S1 Table. The species used in the phylogenetic tree and their Genbank number.**
(DOCX)

**S2 Table. Fossil constraints used in the MCMCtree analyses in this study.**
(DOCX)

**S3 Table. Mitochondrial genomes substitution rates in Phaeophyceae.**
(DOCX)

**S4 Table. Chloroplast genomes substitution rates in Phaeophyceae.**
(DOCX)

**S1 Fig. Venn diagram comparing the protein-coding gene contents of nine brown algal cp-genomes.** The numbers in the Venn diagram represent the number of shared and/or unique gene.
(TIF)

**S2 Fig. Phylogenetic relationship of 19 species in Phaeophyceae inferred from ML and BI analyses based on shared protein-coding genes.** The numbers near each node are bootstrap support values in ML and posterior probability in BI with *H. akashiwo* as outgroup.
(TIF)

**S3 Fig. The synteny analysis of Phaeophyceae mitochondrial genomes.**
(TIF)

## Acknowledgments

We thank Dr. SUN Zhongmin, Miss SONG Xiaohan, and Dr. WANG Jing for providing assistance during field collection and specimens' preservation in Korea.

## Author Contributions

**Investigation:** Han-Gil Choi.

**Resources:** Han-Gil Choi.

**Software:** Shuangshuang Zhang.

**Writing – original draft:** Yanshuo Liang.

**Writing – review & editing:** Zi-Min Hu, Delin Duan.

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
