## [Decision Letter · Decision Letter 0]

19 Jan 2022

PONE-D-21-32930The organellar genomes of Silvetia siliquosa (Fucales, Phaeophyceae) and comparative analyses of the brown algaePLOS ONE

Dear Dr. Duan,

Thank you for submitting your manuscript to PLOS ONE. After careful consideration, we feel that it has merit but does not fully meet PLOS ONE’s publication criteria as it currently stands. Therefore, we invite you to submit a revised version of the manuscript that addresses the points raised during the review process.

We look forward to receiving your revised manuscript.

Kind regards,

Genlou Sun

Academic Editor

PLOS ONE

Journal Requirements:

2. We note that this study contains the sampling of an endangered species. In your Methods section, please provide additional information regarding the permits you obtained for the work. Please ensure you have included the full name of the authority that approved the field site access and, if no permits were required, a brief statement explaining why.

This research were supported by the National Natural Science Foundation of China (Nos. 31971395, 41761144057).

NO

Reviewers' comments:

Reviewer's Responses to Questions

**Comments to the Author**

1. Is the manuscript technically sound, and do the data support the conclusions?

Reviewer #1: Yes

Reviewer #2: Partly

2. Has the statistical analysis been performed appropriately and rigorously? 

Reviewer #1: Yes

Reviewer #2: Yes

3. Have the authors made all data underlying the findings in their manuscript fully available?

Reviewer #1: Yes

Reviewer #2: Yes

4. Is the manuscript presented in an intelligible fashion and written in standard English?

Reviewer #1: Yes

Reviewer #2: Yes

5. Review Comments to the Author

Reviewer #1: Dear Authors,

I have carefully analysed the enclosed manuscript, and my suggestion to the Editor would be to accept your work for publication in PLOS One with minor revision. I believe that you provided a compelling background for your study, performed all the analyses using, to my knowledge, the best available molecular and computational tools, analysed the data very thoroughly and described the findings in a concise, straight-to-the point manner. I am convinced that this publication is definitely important for the current studies of genomics and evolutionary biology of algae. Although this work might not be a major breakthrough overthrowing the paradigms of the field, it fills a substantial void in our knowledge, especially about the mechanisms driving the evolution of organellar genomes and rates of evolution and speciation of non-model, but ecologically and economically significant organisms, such as brown algae.

Below, I have provided a number of minor comments, questions and remarks – most of them are non-scientific/editorial, however, I would be grateful for your response.

Main comments/questions:

Line 115: Why was this particular set of genes chosen for molecular dating? Is it because they are slow-evolving and well conserved across diverse taxa, or are they simply the ones available for most taxa, as indicated later in Line 239? I believe it would be clearer if the reason behind the gene set selection was stated clearly in the Materials and Methods section.

Line 146: The taxonomic name Fucophycidae is used here and later in line 152, however, there is no explanation for what is included in this taxon anywhere in the text or figures. I would suggest that a sentence explaining what exactly are the Fucophycidae should be added at some point in the Introduction, or, alternatively, this taxon could simply be marked on Figure 3 along with order names.

Lines 149-155: This fragment suggests that there are three pairs of overlapping genes in S. siliquosa; however, in Table 1, S. siliquosa is listed as having 10 such pairs. Why are only three of them described in detail in text? Are they the only ones conserved across Fucophycidae, while all others are genus or species-specific?

Lines 212-217: More a comment than question, but in case of the missing chloroplast genes, studies of genomic and transcriptomic data from other algal lineages (e.g. Pelagophyceae – Ong et al. 2010, doi:10.1111/j.1529-8817.2010.00841x, or Dictyochophyceae – Han et al. 2019, doi:10.1111/jpy.12904) have shown that chloroplast-to-nuclear genome transfer is rather frequent among ochrophytes, compared to total gene loss. Of course, a transcriptomic analysis would be far beyond the scope of this study (and not in good taste to request in a review), but I believe it would be worth mentioning that transfer to nucleus is the most plausible explanation for the variation in gene content in brown algal cp-genomes, especially considering that two of the genes missing in certain Phaeophyceae – specifically syfB and rpl32 – have been found to be transferred to the nucleus in some of the Dictyochophyceae (see Han et al. above).

Minor (non-scientific) remarks:

Line 45: …accurate and deep insight (“deep” is twice in the manuscript)

Line 94: … progressive Mauve alignment algorithm (“the” after Mauve is redundant)

Line 134: in “… in Tables 1 and 2 for this research”, I believe “analysis” would be a better-fitting word than “research”, which is more general

Line 145: I believe “S. horneri” should be listed as “Sargassum horneri”, as this particular species has not been referred to before in the text – especially with the prior use of “S.” as Silvetia in the same sentence

Line 148: “…containing 94.31% of known brown algal genes and ORFs” – this phrase suggests that 94.31% of genes that are known from other brown algae is found in S. siliquosa – I believe you mean that 94.31% of its length are coding sequences – known brown algal genes and ORFs?

Line 148: “…non-coding genes” – did you mean non-coding regions?

Lines 231-232: “The reconstructed phylogenetic tree …both formed one sister clade” – I believe it would be better to rephrase it as “…both together formed one clade”, or to make it even simpler - “…supported Laminariales and Ectocarpales as sister monophyletic groups”.

Line 262: I would suggest using “brown algae” instead of Phaeophyta, as this name is not used anywhere else in the text, and even being a synonym of Phaeophyceae, it might just be distracting to a reader not proficient in taxonomy.

Line 287: “volvocine” should not be capitalized, as it is not a Latin name

Line 302: I believe the sentence ends after “…and S. thunbergii”, but the full stop is missing

Line 351: “…higher than that of E. siliculosus” – do you mean “higher than between any of these groups and E. siliculosus”?

Line 357: “…supported in the Sphacelariales” – although it is not highly relevant to the work, I would suggest adding a hint at what the Sphacelariales are, as this name is not mentioned elsewhere in the text (e.g. “…supported in another phaeophycean order Sphacelariales”)

Reviewer #2: Review for PLoS ONE; PONE-D-21-32930

Decision: Reject.

General comments: In this paper, De Duan et al. present the organellar genomes of the fucoidal species Silvetia siliquosa, which is endemic to the Northwest Pacific. The authors extend their analysis to other members of brown seaweed, analysing gene content, genome architecture, sequence evolution, and time calibrated phylogenetic placement. While the organellar genomes of Silvetia are novel to the field, the analyses presented here have already been published with more taxonomically inclusive datasets. I therefore rejected this paper on the grounds that the information presented here is not a substantial step forward for the field. Moreover, several persistent errors regarding gene content are perpetuated in this manuscript.

The first analysis of brown algal organellar genomes with reasonable taxonomic scope was presented by Graf et al. 2017 (PLoS ONE), followed by Liu et al, 2019 (Journal of Molecular Evolution), the latter of which included earlier diverging orders (Ishigeales and Dictyotales), thereby substantially improving our understanding of organellar genome evolution in brown algae. Starko et al., 2021, (Genome Biology and Evolution) is the most recent addition to this arena of analyses, and also improves the taxonomic scope and comprehensiveness of these analyses by publishing novel genomes in 27 species (including from new orders such as Sphacelariales, Desmarestiales, Ralfsiales, and Chordales). Publications presenting organellar genomes from one novel species are simply not justifiable anymore.

The analyses presented here are largely repeated from the work of Starko et al. (2021), but include less taxa, and are therefore arguably less reliable. Nonetheless, the authors arrive at largely the same conclusions presented in earlier publications. Gene content is conserved, as is architecture in mitochondrial genomes, while chloroplast genomes evolve more slowly but gene arrangement varies greatly (but is generally conserved at the ordinal level). Unfortunately, several errors are reinforced in this manuscript. In particular, the authors claim Fucales are missing ycf17, and cite previous claims that petL and ycf54 are missing from Laminariales, and that ycf37 has been lost in Laminaria solidungula with implications on photosynthetic performance in this Arctic species. None of these claims are true, and are easily verifiable by mapping the putatively lost genes to published genomes (I did this exercise myself in geneious while reviewing this manuscript). Here is what Starko et al. wrote on this topic:

“We found that some previously reported cases of gene loss in brown algal plastomes appear to be the result of annotation errors. For example, ycf17 was identified as present in Fucales and ycf54 and petL were found in Laminariales and Chordales contrary to the interpretation of Graf et al. (2017). Moreover, the putative pseudogenization of ycf37 in Laminaria solidungula (Laminariales) reported by Rana et al. (2019) appears to be the result of incorrectly interpreting the fragmented portion of this gene that occurs in one of the inverted repeat regions (while the intact gene straddles the other inverted repeat region).”

These are simply annotation errors that could have been corrected with some very basic QC measures. To the credit of the authors, I was able to confirm the observations on overall gene content reported in the tables, and more specific claims such as the loss of trnL intron in Ectocarpales.

The only novel analysis presented here is the time calibrated phylogeny, but I question the rigour of these results as the authors did not include other published genomes from Chordales, Ralfsiales, Desmarestiales, Sphacelariales, and Ishegeales. Overall, as the manuscript stands, I think the work presented here has the potential to add considerable confusion to the literature.

My recommendation is to either sequence several new organellar genomes to improve on existing work (thereby justifying this work), or sequence Silvetia to a greater depth and pursue knowledge of the nuclear genome (for which little is published in brown algae). I’m sorry my review is not more positive.

Specific comments:

43: I do not understand how information on organellar genomes are linked to conservation efforts. Besides offering greater access to variant positions, they offer little to no relevant functional performance information (in the brown seaweeds at least, because gene content is conserved).

53-55: see general comments on why this statement is not true. Ideally, the Rana et al. study should be retracted, as this claim was the basis and main conclusion for the paper.

55-58: We already know gene content is conserved, undermining this statement. As well, organellar genomes are fairly well characterized, with scope now capturing the breadth of Phaeophyceae since the starko et al. 2021 study. Evidently, however, organellar genomes remain to be sequenced in several less well studied orders.

80: The authors here a priori assume organellar topology consistent with other species of Fucales. Looking at the genomes, clearly the authors correctly assembled the genomes. My recommendation for future work, however, is to employ different assembly methods to arrive at answers without any a priori assumptions made. For instance, the authors can elongate a seed sequence using NOVOPlasty, which confirms circularity, and should ensure to map reads back to the genome to ensure assembly errors are not present (i.e. coverage should be consistent, with no breaks in overlapping sequences).

104: the authors should include other published genomes from other orders, see my general comment above. Desmarestia appears in Table S1, but why not in the phylogenetic tree

204-209: see comment above on genes that are incorrectly reported to be absent

333-334: many more rearrangements depicted by Starko et al. 2021

6. PLOS authors have the option to publish the peer review history of their article (what does this mean?). If published, this will include your full peer review and any attached files.

Reviewer #1: No

Reviewer #2: No

---

## [Author Response · Author response to Decision Letter 0]

19 Feb 2022

To Reviewer #1:

Main comments/questions:

Question 1: Line 115: Why was this particular set of genes chosen for molecular dating? Is it because they are slow-evolving and well conserved across diverse taxa, or are they simply the ones available for most taxa, as indicated later in Line 239? I believe it would be clearer if the reason behind the gene set selection was stated clearly in the Materials and Methods section.

Response: Thanks for this valuable comment. We selected 15 taxa with both mitochondrial and chloroplast genomes to be sequenced. These genes (cox1, cox3, nad1, nad4, atp9, rbcL, psbA, and atpB) are used to molecular dating due to their high conservation and slow evolutionary rate among brown algal taxa, and they are widely used in the estimation of divergence time in brown algae (Silberfeld et al., 2010; Starko et al., 2019). As suggested, we added instructions in the materials and methods as “These genes were highly conserved and slow-evolving”.

Reference：

Silberfeld T, Leigh JW, Verbruggen H, Cruaud C, Reviers BD, Rousseau F. A multi-locus time-calibrated phylogeny of the brown algae (Heterokonta, Ochrophyta, Phaeophyceae): Investigating the evolutionary nature of the “brown algal crown radiation”. Mol Phylogenet Evol. 2010; 56: 659−674. doi: 10.1016/j.ympev.2010.04.020

Starko S, Gomez MS, Darby H, Demes KW, Kawai H, Yotsukura N, et al. A comprehensive kelp phylogeny sheds light on the evolution of an ecosystem. Mol Phylogenet Evol. 2019; 136: 138−150. doi: 10.1016/j.ympev.2019.04.012

Question 2: Line 146: The taxonomic name Fucophycidae is used here and later in line 152, however, there is no explanation for what is included in this taxon anywhere in the text or figures. I would suggest that a sentence explaining what exactly are the Fucophycidae should be added at some point in the Introduction, or, alternatively, this taxon could simply be marked on Figure 3 along with order names.

Response: As suggested, we have marked the class names (Fucophycidae and Phaeophyceae) in Figure 3.

Question 3: Lines 149-155: This fragment suggests that there are three pairs of overlapping genes in S. siliquosa; however, in Table 1, S. siliquosa is listed as having 10 such pairs. Why are only three of them described in detail in text? Are they the only ones conserved across Fucophycidae, while all others are genus or species-specific?

Response: Yes, although 10 pairs of overlapping regions were detected in the S. siliquosa mitochondrial genome, these three pairs of overlapping regions were found to be highly conserved among Fucophycidae species by comparison with other brown algae.

Question 4: Lines 212-217: More a comment than question, but in case of the missing chloroplast genes, studies of genomic and transcriptomic data from other algal lineages (e.g. Pelagophyceae – Ong et al. 2010, doi:10.1111/j.1529-8817.2010.00841x, or Dictyochophyceae – Han et al. 2019, doi:10.1111/jpy.12904) have shown that chloroplast-to-nuclear genome transfer is rather frequent among ochrophytes, compared to total gene loss. Of course, a transcriptomic analysis would be far beyond the scope of this study (and not in good taste to request in a review), but I believe it would be worth mentioning that transfer to nucleus is the most plausible explanation for the variation in gene content in brown algal cp-genomes, especially considering that two of the genes missing in certain Phaeophyceae – specifically syfB and rpl32 – have been found to be transferred to the nucleus in some of the Dictyochophyceae (see Han et al. above).

Response: Thanks for this constructive comment. Horizontal gene transfer is usually determined based on homologous sequence alignment analysis. Currently, there are few available nuclear genome data for brown algae, so it is difficult to compare the homology between chloroplast and nuclear genomes. We also believe that with increasing genomic data, the veil of horizontal gene transfer events between organelles and nuclear genomes of brown algae will be uncovered.

Minor (non-scientific) remarks:

Question 1：Line 45: …accurate and deep insight (“deep” is twice in the manuscript)

Response: As suggested, we have deleted the extra word “deep”.

Question 2：Line 94: … progressive Mauve alignment algorithm (“the” after Mauve is redundant)

Response: As suggested, we have deleted the word “the”.

Question 3：Line 134: in “… in Tables 1 and 2 for this research”, I believe “analysis” would be a better-fitting word than “research”, which is more general

Response: Thanks, we changed the word “research” to “analysis”.

Question 4：Line 145: I believe “S. horneri” should be listed as “Sargassum horneri”, as this particular species has not been referred to before in the text – especially with the prior use of “S.” as Silvetia in the same sentence

Response: As suggested, we replaced ‘S. horneri” by using “Sargassum horneri.”

Question 5：Line 148: “…containing 94.31% of known brown algal genes and ORFs” – this phrase suggests that 94.31% of genes that are known from other brown algae is found in S. siliquosa – I believe you mean that 94.31% of its length are coding sequences – known brown algal genes and ORFs?

Response: Yes, we didn't make it clear. Accordingly, this sentence has been rephrased to “The mt-genome of S. siliquosa is gene dense, and the length of coding genes accounts for 94.31% of the total mt-genome, and non-coding regions accounts for only 5.69%, well within the range of Phaeophycean species (3.21-6.49%, Table 1).”

Question 6：Line 148: “…non-coding genes” – did you mean non-coding regions?

Response: Yes, as suggested, we have changed “non-coding genes” to “non-coding regions.”

Question 7：Lines 231-232: “The reconstructed phylogenetic tree …both formed one sister clade” – I believe it would be better to rephrase it as “…both together formed one clade”, or to make it even simpler - “…supported Laminariales and Ectocarpales as sister monophyletic groups.”

Response: As suggested, this sentence has been rephrased to “The reconstructed phylogenetic tree supported Laminariales and Ectocarpales as sister monophyletic groups.”

Question 8：Line 262: I would suggest using “brown algae” instead of Phaeophyta, as this name is not used anywhere else in the text, and even being a synonym of Phaeophyceae, it might just be distracting to a reader not proficient in taxonomy.

Response: Thanks, we changed “brown algae” to “Phaeophyceae species”.

Question 9：Line 287: “volvocine” should not be capitalized, as it is not a Latin name

Response: Thanks for this constructive comment, accordingly we changed “volvocine” to “Volvulina compacta”.

Question 10：Line 302: I believe the sentence ends after “…and S. thunbergii”, but the full stop is missing

Response: Thanks, we have modified it.

Question 11：Line 351: “…higher than that of E. siliculosus” – do you mean “higher than between any of these groups and E. siliculosus”?

Response: Yes, we have rephrased it accordingly.

Question 12：Line 357: “…supported in the Sphacelariales” – although it is not highly relevant to the work, I would suggest adding a hint at what the Sphacelariales are, as this name is not mentioned elsewhere in the text (e.g. “…supported in another phaeophycean order Sphacelariales”)

Response: Thanks for this comment. We have rephrased it accordingly.

To Reviewer #2:

Specific comments:

Question 1：line 43: I do not understand how information on organellar genomes are linked to conservation efforts. Besides offering greater access to variant positions, they offer little to no relevant functional performance information (in the brown seaweeds at least, because gene content is conserved).

Response: Thanks, although our study was not effective for the proliferation of this threatened algae, as a method of ex situ conservation, establishing gene bank of threatened algae can provide basic research data for the conservation of threatened algae.

Question 2：line 43: 53-55: see general comments on why this statement is not true. Ideally, the Rana et al. study should be retracted, as this claim was the basis and main conclusion for the paper.

Response: Thanks for this valuable comment, accordingly we deleted this citation and reference.

Question 3：line 55-58: We already know gene content is conserved, undermining this statement. As well, organellar genomes are fairly well characterized, with scope now capturing the breadth of Phaeophyceae since the starko et al. 2021 study. Evidently, however, organellar genomes remain to be sequenced in several less well studied orders.

Response: Thanks for this comment. We have rephrased it accordingly as follows: 

“For example, designing molecular markers based on gene variable regions (nucleotide insertion/deletion) can be used for species identification [10, 17]. However, there is still limited genomic information in the brown alga, limiting our understanding of the taxonomic status and evolutionary history of the Phaeophyceae.”

Reference:

10. Yotsukura N, Shimizu T, Katayama T, Druehl LD. Mitochondrial DNA sequence variation of four Saccharina species (Laminariales, Phaeophyceae) growing in Japan. J Appl Phycol. 2010; 22: 243−251. doi: 10.1007/s10811-009-9452-7

17. Melton JT, Leliaert F, Tronholm A, Lopez-Bautista JM. The complete chloroplast and mitochondrial genomes of the green macroalga Ulva sp. UNA00071828 (Ulvophyceae, Chlorophyta). PLoS ONE. 2019; 10: e0121020. doi: 10.1371/journal.pone.0121020

Question 4：line 80: The authors here a priori assume organellar topology consistent with other species of Fucales. Looking at the genomes, clearly the authors correctly assembled the genomes. My recommendation for future work, however, is to employ different assembly methods to arrive at answers without any a priori assumptions made. For instance, the authors can elongate a seed sequence using NOVOPlasty, which confirms circularity, and should ensure to map reads back to the genome to ensure assembly errors are not present (i.e. coverage should be consistent, with no breaks in overlapping sequences).

Response: Thanks for this constructive comment.

Question 5：line 104: the authors should include other published genomes from other orders, see my general comment above. Desmarestia appears in Table S1, but why not in the phylogenetic tree

Response: Thanks, the phylogenetic tree containing Desmarestia can be found in Fig. S2.

Question 6：line 204-209: see comment above on genes that are incorrectly reported to be absent

Response: Thanks for your constructive comment. Starko et al. (2021) suggested that the loss of ycf17 gene appear to be the result of annotation errors, but we used ycf17 in Saccharina japonica (Genbank number: NC_013476) for homology comparison based on Nucleotide Blast, and this homologous gene was not found in the chloroplast genome of the reported order Fucales. In addition, how to find this gene in Fucales was not explained in Starko's article. None of the previously published chloroplast genomes of Fucales contain ycf17 gene (Le Corguillé et al., 2009; Liu and Pang, 2016; Yang et al., 2016; Graf et al., 2017; Liu et al., 2018).

Reference:

Le Corguillé G, Pearson G, Valente M, Viegas C, Gschloessl B, Corre E, et al. Plastid genomes of two brown algae, Ectocarpus siliculosus and Fucus vesiculosus: further insights on the evolution of red-algal derived plastids. BMC Evol Biol. 2009; 9: 253. doi: 10.1186/1471-2148-9-253

Liu and Pang. Chloroplast genome of Sargassum horneri (Sargassaceae, Phaeophyceae): comparative chloroplast genomics of brown algae. J Appl Phycol. 2016; 28: 1419−1426. doi: 10.1007/s10811-015-0609-2

Yang JH, Graf L, Cho CH, Jeon BH, Kim JH, Yoon HS. Complete plastid genome of an ecologically important brown alga Sargassum thunbergii (Fucales, Phaeophyceae). Mar Genomics. 2016; 28: 17−20. doi: 10.1016/j.margen.2016.03.003

Graf L, Kim YJ, Cho GY, Miller KA, Yoon HS. Plastid and mitochondrial genomes of Coccophora langsdorfii (Fucales, Phaeophyceae) and the utility of molecular markers. PLoS ONE. 2017; 12: e0187104. doi: 10.1371/journal.pone.0187104

Liu F, Pan J, Zhang ZS, Moejes FW. Organelle genomes of Sargassum confusum (Fucales, Phaeophyceae): mtDNA vs cpDNA. J Appl Phycol. 2018; 30: 2715−2722. doi: 10.1007/s10811-018-1461-y

Question 7：line 333-334: many more rearrangements depicted by Starko et al. 2021

Response: Sorry, we did not understand your question. Do you mean that the results here are wrong? However, in lines 333 to 334, we found rearrangement of four genes (atp8, rpl31, rps10, and atp9) in the mitochondrial genome of D. dichotoma, which is consistent with Starko's results (Starko et al., 2021; Fig. 4).

Reference:

Starko S, Bringloe TT, Gomez MS, Darby H, Graham SW, Martone PT. Genomic rearrangements and sequence evolution across brown algal organelles. Genome Biol Evol. 2021; 7: evab124. doi: 10.1093/gbe/evab124/6290714

---

## [Decision Letter · Decision Letter 1]

4 Mar 2022

PONE-D-21-32930R1The organellar genomes of Silvetia siliquosa (Fucales, Phaeophyceae) and comparative analyses of the brown algaePLOS ONE

Dear Dr. Duan,

Thank you for submitting your manuscript to PLOS ONE. After careful consideration, we feel that it has merit but does not fully meet PLOS ONE’s publication criteria as it currently stands. Therefore, we invite you to submit a revised version of the manuscript that addresses the points raised during the review process.

 Please address the concerns raised by the reviewer #2.

We look forward to receiving your revised manuscript.

Kind regards,

Genlou Sun

Academic Editor

PLOS ONE

Additional Editor Comments:

Please address the concerns raised by reviewer #2.

Reviewers' comments:

Reviewer's Responses to Questions

**Comments to the Author**

1. If the authors have adequately addressed your comments raised in a previous round of review and you feel that this manuscript is now acceptable for publication, you may indicate that here to bypass the “Comments to the Author” section, enter your conflict of interest statement in the “Confidential to Editor” section, and submit your "Accept" recommendation.

Reviewer #1: All comments have been addressed

Reviewer #2: (No Response)

2. Is the manuscript technically sound, and do the data support the conclusions?

Reviewer #1: Yes

Reviewer #2: No

3. Has the statistical analysis been performed appropriately and rigorously? 

Reviewer #1: Yes

Reviewer #2: No

4. Have the authors made all data underlying the findings in their manuscript fully available?

Reviewer #1: Yes

Reviewer #2: No

5. Is the manuscript presented in an intelligible fashion and written in standard English?

Reviewer #1: Yes

Reviewer #2: Yes

6. Review Comments to the Author

Reviewer #1: Dear Authors,

I believe all the issues have been addressed in a satisfactory manner - therefore, I would recommend for this work to be accepted for publication. There is just a single misspelled name I noticed (Line 302 - "S. thunbergia" instead of "thunbergii"), but it can be corrected in the proof. Otherwise, I have no further remarks - good work and good luck in your future scientific endeavours!

Best regards

Reviewer #2: Review for PLoS ONE; PONE-D-21-32930R1

Decision: Reject.

General comments: In this paper, De Duan et al. present the organellar genomes of the fucoidal species Silvetia siliquosa, which is endemic to the Northwest Pacific. The authors extend their analysis to other members of brown seaweed, analysing gene content, genome architecture, sequence evolution, and time calibrated phylogenetic placement. In my initial review, I pointed out several errors in interpreting putatively lost genes, and argued that many of the findings were already presented in more taxonomically inclusive datasets. Unfortunately, my initial analyses still stands in this revised manuscript, as the authors have not adequately addressed the concerns I raised, hence my decision to uphold a rejection. Before I can endorse the paper, I’d appreciate detailed responses to the following points:

1) The phylogenetic analyses are not taxonomically inclusive. What is the justification from the authors for not including other published genomes in their analyses, particularly from the orders Chordales (mito: MZ156045, MZ156050, MZ156063; plastid: MZ156027, MZ156037, MZ156030), Sphacelariales (mito: MZ156064; plastid: MZ156028), Ralfsiales (mito: MZ156065; plastid: NA); and Ishegeales (mito: MG940857; plastid: NA). I cannot think of a justifiable reason to not use all the available data for these inferences.

2) The authors continue to reiterate gene loss information that I don’t believe is supported by their data. Unfortunately, as I cannot access the sequences for Silvetia, I cannot verify this. Taking the authors phylogenetic tree of mitochondrial markers, I assume Fucus is the closest relative with a published genome to verify their claims. Taking Fucus vesiculosus MG922855 then, one can take the ycf17 gene from another brown alga (I arbitrarily did this for Postelsia palmaeformis MZ156031) and map this sequence to MG922855 using built in geneious mapping algorithm or bbmap plugin. A gene sequence region is revealed positioned just before ycf19, with the start and stop positions slightly shifted compared to postelsia.

>ycf17_fucus_vesiculosus_ MG922855 (reversed)

ATGAATAAAGACTACAATTATAACAATTCTATAGAGTATAAAATAAGATGGGGGTTCTATTTAAAAAATGAAATTTTAAATGGTCGTGGTGCAATGATTTTATTAATAATAATAATATTATTAGAAATTTTTACACATAAAACTATAGTAAATTTAATCTTTCAAAGGTAA

You can then submit the translated sequence to interproscan (https://www.ebi.ac.uk/interpro/search/sequence/) and you will see both sequences from Fucus and Postelsia, while not predicted to be a specific gene, has nearly identical domains, one for chlorophyll a-b binding. The authors can also blast the translated sequence and see it matches ycf17 in other brown algae (a nucleotide blast does not come up with matches). Or, the authors can take ycf17 annotations from other brown seaweeds and align it with the above sequence. They will see the above sequence is indeed ycf17.

Similar exercises can be carried out in the other putatively lost genes, including ycf54 and petL for other orders such as Laminariales. The authors cite several studies publishing Fucales organellar genomes without ycf17; annotation algorithms are not fool proof, and we are prone to perpetuating errors if left unchecked. The authors can verify the presence of ycf17 in other fucales; Coccophora has a similar above sequence, also positioned prior to ycf19. Unless the authors have a compelling rebuttal, which I am of course open to, I must insist they do not continue to perpetuate errors in the literature regarding these lost genes.

3) The authors are very descriptive in the discussion without providing any context for the observations, e.g. 145-193. Are these observations typical? What, if anything, does this tell us about the biology of brown seaweeds. If there is no bigger picture, what then is the relevance of these results? In particular, the authors missed an opportunity to describe divergence within the context of earth’s history, both in terms of climate and biome, or how evolutionary histories in different lineages shaped substitution rates.

4) Please also upload your short read data to the short read archive, otherwise the genome assembly, the foundation of this work, is not verifiable or replicable.

I’d appreciate it if the authors could clarify points of confusion in the text indicated in my specific comments.

Specific comments:

48: Recent whole genome work calls into question this assumption of using organellar genomes for phylogenetic inferences. Phylogenetic inferences are hampered by uniparentally inherited genomes (i.e. organelles) in the presence of hybridization and organellar capture, which appears to the case in some brown algae (Bringloe et al. 2021, Journal of Phycology, Whole genome sequencing reveals..). In this case of the kelp Alaria, each genomic compartment (i.e. mitochondrial, plastid, and nuclear) revealed completely different topologies and evolutionary histories. It might be the case that relationships in closely related species of brown algae are not accurately reflected in organellar genomes if hybridizations is common, something we cannot easily evaluate or detect without whole genome data.

52: what is meant here by “gene variable regions” and how does this relate to insertions or deletions, which are (as far as I know) extremely rare within genes in brown algal organellar genomes (since this would cause the gene to be non functional by shifting the reading frame)

54: Can the authors be more specific here. There is good coverage in fucales and Laminariales, but relatively little in less well studied brown algal orders. There are a lot of published organellar genomes, but they are severely taxonomically biased.

59-60: what is meant by typical brown algae?

173-179: Here and elsewhere in the discussion, the material is very descriptive. If presenting this information, can the authors clarify its relevance? Are these observations abnormal or species compared to other species, brown algae or otherwise? I bigger picture is needed.

205-210: see above comment on why ycf17 is indeed present in Fucales, and petL and ycf54 are present in laminariales.

332: speaking to my previous comment, which the authors asked for clarification on, I would dispute the claim that mitochondrial genome architecture is highly conserved. One rearrangement is cited in the discussion, however, many are depicted in Starko et al 2021. Again, the constrained taxonomic scope of this paper is misleading.

357-359: Not clear how rearrangements play a role in species divergence, can the authors explain this a bit further. Otherwise this comes across as wild speculation. If there is a compelling hypothesis here, it would be helpful for the authors to be clear on what is occurring and how it might be validated in future work.

Fig.6: The long and short single copy regions are out of order for Ectocarpus, making this figure harder to interpret in terms of smaller rearrangements (the figure implies the short and large single copy regions are inverted in this species). The authors should rearrange the figure so the long single copy region appears first in Ectocarpus, consistent with the other species depicted here.

What is the basis for the taxa selected in this figure? Why are other published arrangements not included? E.g. Sphacelariales, Desmarestiales, ect.

7. PLOS authors have the option to publish the peer review history of their article (what does this mean?). If published, this will include your full peer review and any attached files.

Reviewer #1: No

Reviewer #2: No

---

## [Author Response · Author response to Decision Letter 1]

25 Mar 2022

To Reviewer #1:

Question 1: Line 302 - "S. thunbergia" instead of "thunbergii"

Response: Thanks, we have modified it.

To Reviewer #2:

General comments

Question 1: The phylogenetic analyses are not taxonomically inclusive. What is the justification from the authors for not including other published genomes in their analyses, particularly from the orders Chordales (mito: MZ156045, MZ156050, MZ156063; plastid: MZ156027, MZ156037, MZ156030), Sphacelariales (mito: MZ156064; plastid: MZ156028), Ralfsiales (mito: MZ156065; plastid: NA); and Ishegeales (mito: MG940857; plastid: NA). I cannot think of a justifiable reason to not use all the available data for these inferences.

Response: The purpose of our phylogenetic tree is not to solve the evolutionary relationship between brown algae, but to solve the taxonomic status of S. siliquosa in brown algae, so we did not add additional taxa. Phylogenetic analysis revealed a close genetic relationship between S. siliquosa and F. vesiculosus, which diverged approximately 8 Mya (5.7–11.0 Mya), corresponding to the Late Miocene (5.3–11.6 Ma).

Question 2: The authors continue to reiterate gene loss information that I don’t believe is supported by their data. Unfortunately, as I cannot access the sequences for Silvetia, I cannot verify this. Taking the authors phylogenetic tree of mitochondrial markers, I assume Fucus is the closest relative with a published genome to verify their claims. Taking Fucus vesiculosus MG922855 then, one can take the ycf17 gene from another brown alga (I arbitrarily did this for Postelsia palmaeformis MZ156031) and map this sequence to MG922855 using built in geneious mapping algorithm or bbmap plugin. A gene sequence region is revealed positioned just before ycf19, with the start and stop positions slightly shifted compared to postelsia.

>ycf17_fucus_vesiculosus_ MG922855 (reversed)

ATGAATAAAGACTACAATTATAACAATTCTATAGAGTATAAAATAAGATGGGGGTTCTATTTAAAAAATGAAATTTTAAATGGTCGTGGTGCAATGATTTTATTAATAATAATAATATTATTAGAAATTTTTACACATAAAACTATAGTAAATTTAATCTTTCAAAGGTAA

You can then submit the translated sequence to interproscan (https://www.ebi.ac.uk/interpro/search/sequence/) and you will see both sequences from Fucus and Postelsia, while not predicted to be a specific gene, has nearly identical domains, one for chlorophyll a-b binding. The authors can also blast the translated sequence and see it matches ycf17 in other brown algae (a nucleotide blast does not come up with matches). Or, the authors can take ycf17 annotations from other brown seaweeds and align it with the above sequence. They will see the above sequence is indeed ycf17.

Similar exercises can be carried out in the other putatively lost genes, including ycf54 and petL for other orders such as Laminariales. The authors cite several studies publishing Fucales organellar genomes without ycf17; annotation algorithms are not fool proof, and we are prone to perpetuating errors if left unchecked. The authors can verify the presence of ycf17 in other fucales; Coccophora has a similar above sequence, also positioned prior to ycf19. Unless the authors have a compelling rebuttal, which I am of course open to, I must insist they do not continue to perpetuate errors in the literature regarding these lost genes.

Response: Thanks for this valuable comment. As suggested, we re-annotated the ycf17 gene in the S. siliquosa chloroplast genome and re-uploaded it to NCBI （Genebank number: MW485980）. In addition, we re-analyzed the gene components of the chloroplast genomes of brown algae.

Question 3: The authors are very descriptive in the discussion without providing any context for the observations, e.g. 145-193. Are these observations typical? What, if anything, does this tell us about the biology of brown seaweeds. If there is no bigger picture, what then is the relevance of these results? In particular, the authors missed an opportunity to describe divergence within the context of earth’s history, both in terms of climate and biome, or how evolutionary histories in different lineages shaped substitution rates.

Response: Thanks, we analyzed the gene composition of the mitochondrial and chloroplast genomes of S. siliquosa and compared them with those of other brown algae. However, our results did not find significant novelty, such as gene loss and gain, so we cannot discuss it in a better perspective. Our goal is to fully analyze the organelle genomes of common brown algae including the species we study, and to explore the changes of organelle structure.

Question 4: Please also upload your short read data to the short read archive, otherwise the genome assembly, the foundation of this work, is not verifiable or replicable.

Response: The genome sequences information of our organelles has been uploaded to NCBI, and you can query them by Genebank number MW485976 and MW485980, respectively. For the chloroplast genome, we re-annotated ycf17 according to your suggestion.

Specific comments:

Question 1: 48: Recent whole genome work calls into question this assumption of using organellar genomes for phylogenetic inferences. Phylogenetic inferences are hampered by uniparentally inherited genomes (i.e. organelles) in the presence of hybridization and organellar capture, which appears to the case in some brown algae (Bringloe et al. 2021, Journal of Phycology, Whole genome sequencing reveals..). In this case of the kelp Alaria, each genomic compartment (i.e. mitochondrial, plastid, and nuclear) revealed completely different topologies and evolutionary histories. It might be the case that relationships in closely related species of brown algae are not accurately reflected in organellar genomes if hybridizations is common, something we cannot easily evaluate or detect without whole genome data.

Response: Thanks for this constructive comment. For brown algae, the genomes of only 7 species have been sequenced (https://www.ncbi.nlm.nih.gov/genome/?term=txid2870[Organism:exp]), so it is difficult to construct the tree of life of brown algae. We also admit that the current organelle genome sequencing may not reflect the evolutionary history and topological relationship between brown algae, but for now, the construction of evolutionary tree using organelle genome is the most effective methods to solve the evolutionary relationship. In addition, the sequencing of S. siliquosa genome is ongoing, hoping to better solve the evolutionary relationship of brown algae in the future. We have rephrased it accordingly as follows:

“Complete organelle genome data can provide important reference for the phylogenetic construction of brown algae.”

Question 2: 52: what is meant here by “gene variable regions” and how does this relate to insertions or deletions, which are (as far as I know) extremely rare within genes in brown algal organellar genomes (since this would cause the gene to be non functional by shifting the reading frame).

Response: Thanks, we have rephrased it accordingly as follows:

“For example, designing molecular markers based on polymorphism can be used for species identification.”

Question 3: 54: Can the authors be more specific here. There is good coverage in fucales and Laminariales, but relatively little in less well studied brown algal orders. There are a lot of published organellar genomes, but they are severely taxonomically biased.

Response: Thanks, we have rephrased it accordingly as follows:

“Additional organelle genomes from novel taxa will not only provides data support for analyzing the structural variation of organelle genomes, but also advance our understanding of the evolution and diversity of brown algae.”

Question 4: 59-60: what is meant by typical brown algae?

Response: Typical brown algae refers to the common economic brown algae.

Question 5: 173-179: Here and elsewhere in the discussion, the material is very descriptive. If presenting this information, can the authors clarify its relevance? Are these observations abnormal or species compared to other species, brown algae or otherwise? I bigger picture is needed.

Response: Here, we would like to show the usage of start and stop codons in the mitochondrial genome of S. siliquosa, and compare it with other brown algae, and would like to present a comparative result to the reader, as the title of our paper also mentions "comparative analyses of the brown algae”.

Question 6: 205-210: see above comment on why ycf17 is indeed present in Fucales, and petL and ycf54 are present in laminariales.

Response: Thanks for this valuable comment, accordingly we have re-annotated the chloroplast genome of S. siliquosa, and this part of the content has been modified.

Question 7: 332: speaking to my previous comment, which the authors asked for clarification on, I would dispute the claim that mitochondrial genome architecture is highly conserved. One rearrangement is cited in the discussion, however, many are depicted in Starko et al 2021. Again, the constrained taxonomic scope of this paper is misleading.

Response: Thanks for this constructive comment, as you said, fewer taxa were used in the analysis of mitochondrial genome architecture and only one rearrangement was found, but there was nothing wrong with our results. Furthermore, our final conclusion is to illustrate that the structural variation of the chloroplast genome of the brown algae is significantly larger than that of the mitochondrial genome, which is consistent with the results of Starko et al (2021). We have rephrased it accordingly.

Question 8: 357-359: Not clear how rearrangements play a role in species divergence, can the authors explain this a bit further. Otherwise this comes across as wild speculation. If there is a compelling hypothesis here, it would be helpful for the authors to be clear on what is occurring and how it might be validated in future work.

Response: Thanks, this hypothesis was proposed because, based on rearrangement data, we found that brown algal chloroplast genomes exhibited many rearrangement and inversion events at the different order level, with less structural variation within the same order.

Question 9: Fig.6: The long and short single copy regions are out of order for Ectocarpus, making this figure harder to interpret in terms of smaller rearrangements (the figure implies the short and large single copy regions are inverted in this species). The authors should rearrange the figure so the long single copy region appears first in Ectocarpus, consistent with the other species depicted here.

What is the basis for the taxa selected in this figure? Why are other published arrangements not included? E.g. Sphacelariales, Desmarestiales, ect.

Response: Figure 6 has been modified as suggested. In addition, the taxa were chosen for the sole purpose of supporting or validating our results, and we found that the current data set is sufficient to support the conclusion that the chloroplast genomes of brown algae are more structurally variable than the mitochondrial genomes.

---

## [Decision Letter · Decision Letter 2]

7 Apr 2022

PONE-D-21-32930R2The organellar genomes of Silvetia siliquosa (Fucales, Phaeophyceae) and comparative analyses of the brown algaePLOS ONE

Dear Dr. Duan,

Thank you for submitting your manuscript to PLOS ONE. After careful consideration, we feel that it has merit but does not fully meet PLOS ONE’s publication criteria as it currently stands. Therefore, we invite you to submit a revised version of the manuscript that addresses the points raised during the review process.

One of the reviewers recommended rejection, but with open of acceptance after revision. The reviewer commend the authors on making corrections regarding gene losses, and insist (again) that you upload their short read data to the short read archive, in addition to your submission of the fully assembled genomes to genbank (which you have already done). The assembled genomes are an endpoint in your analysis, not the data you generated, the latter of which must be available for re-analysis. The raw data (the short reads you used to map and assemble the organellar genomes) are fundamental to the repeatability of this work (at least, insofar as it relates to reproducing the organellar genomes of S. siliquosa).

We look forward to receiving your revised manuscript.

Kind regards,

Genlou Sun

Academic Editor

PLOS ONE

Additional Editor Comments:

One of the reviewers suggested rejection, but the reviewer is open for acceptance after revision. Please take the comments into consideration for revision.

Reviewers' comments:

Reviewer's Responses to Questions

**Comments to the Author**

1. If the authors have adequately addressed your comments raised in a previous round of review and you feel that this manuscript is now acceptable for publication, you may indicate that here to bypass the “Comments to the Author” section, enter your conflict of interest statement in the “Confidential to Editor” section, and submit your "Accept" recommendation.

Reviewer #1: All comments have been addressed

Reviewer #2: (No Response)

2. Is the manuscript technically sound, and do the data support the conclusions?

Reviewer #1: Yes

Reviewer #2: Partly

3. Has the statistical analysis been performed appropriately and rigorously? 

Reviewer #1: Yes

Reviewer #2: No

4. Have the authors made all data underlying the findings in their manuscript fully available?

Reviewer #1: Yes

Reviewer #2: No

5. Is the manuscript presented in an intelligible fashion and written in standard English?

Reviewer #1: Yes

Reviewer #2: Yes

6. Review Comments to the Author

Reviewer #1: All comments and suggestions have been addressed. I have submitted my recommendation to accept this work for publication.

Reviewer #2: General comments: In this paper, De Duan et al. present the organellar genomes of the fucoidal species Silvetia siliquosa, which is endemic to the Northwest Pacific. The authors extend their analysis to other members of brown seaweed, analysing gene content, genome architecture, sequence evolution, and time calibrated phylogenetic placement. In my initial review, I pointed out several errors in interpreting putatively lost genes, and argued that many of the findings were already presented in more taxonomically inclusive datasets. In the most recent submission, the errors regarding gene loss have been corrected, but I still have some outstanding issues with the paper.

First, I have remained confused through this review process regarding the taxonomic scope of the paper. The authors insist the paper is about comparing the organellar genomes of S. siliquosa to other brown algae, specifically to resolve its phylogenetic placement (which they used to justify the taxonomic scope of their time calibrated tree), but they also analyse the organellar genomes of most (but not all) representative brown seaweeds with respect to genomic architecture and sequence evolution, and spend most of the discussion elaborating at this broader taxonomic scale. Because the scope of the paper extends beyond S. siliquosa, I remain confused as to why the authors did not draw in all the available genomes for brown seaweeds into their analyses. I have a fundamental problem with an analysis of the phaeophyceae that ignores some of the available data, especially at the ordinal level where differences in patterns are most likely to emerge. While I do not agree with the conflation of these objectives and how the data was gathered and analysed at the broader scale of all brown seaweeds, the editor will ultimately have to decide if this is acceptable for publication. Perhaps a compromise is for the authors to acknowledge in the manuscript that they did not analyse all the available brown algal genomes at the ordinal level (I listed these in my last review). This, at least, would make this aspect of the manuscript transparent to the reader.

I also took issue with the broader implications and interpretations of this work in my previous review. I think this is because the manuscript is very descriptive. I wonder how useful or relevant it is to present information such as total gene length, space content, or overlap sizes, especially when these numbers are so highly conserved across brown seaweeds. I’m just not clear on how or why these details are relevant. But the authors are entitled to present this, and if the editor is fine with a descriptive study of this nature, then I respect that decision.

While I commend the authors on making corrections regarding gene losses, I must insist (again) that they upload their short read data to the short read archive, in addition to their submission of the fully assembled genomes to genbank (which they have already done). The assembled genomes are an endpoint in your analysis, not the data you generated, the latter of which must be available for re-analysis. The raw data (the short reads you used to map and assemble the organellar genomes) are fundamental to the repeatability of this work (at least, insofar as it relates to reproducing the organellar genomes of S. siliquosa). This must be done before I can endorse for publication.

7. PLOS authors have the option to publish the peer review history of their article (what does this mean?). If published, this will include your full peer review and any attached files.

Reviewer #1: No

Reviewer #2: No

---

## [Author Response · Author response to Decision Letter 2]

2 May 2022

To Reviewer #1:

Review Comment: All comments and suggestions have been addressed. I have submitted my recommendation to accept this work for publication.

Response: Thank you for reviewing this article. Thank you very much.

To Reviewer #2:

General comments: In this paper, De Duan et al. present the organellar genomes of the fucoidal species Silvetia siliquosa, which is endemic to the Northwest Pacific. The authors extend their analysis to other members of brown seaweed, analysing gene content, genome architecture, sequence evolution, and time calibrated phylogenetic placement. In my initial review, I pointed out several errors in interpreting putatively lost genes, and argued that many of the findings were already presented in more taxonomically inclusive datasets. In the most recent submission, the errors regarding gene loss have been corrected, but I still have some outstanding issues with the paper.

First, I have remained confused through this review process regarding the taxonomic scope of the paper. The authors insist the paper is about comparing the organellar genomes of S. siliquosa to other brown algae, specifically to resolve its phylogenetic placement (which they used to justify the taxonomic scope of their time calibrated tree), but they also analyse the organellar genomes of most (but not all) representative brown seaweeds with respect to genomic architecture and sequence evolution, and spend most of the discussion elaborating at this broader taxonomic scale. Because the scope of the paper extends beyond S. siliquosa, I remain confused as to why the authors did not draw in all the available genomes for brown seaweeds into their analyses. I have a fundamental problem with an analysis of the phaeophyceae that ignores some of the available data, especially at the ordinal level where differences in patterns are most likely to emerge. While I do not agree with the conflation of these objectives and how the data was gathered and analysed at the broader scale of all brown seaweeds, the editor will ultimately have to decide if this is acceptable for publication. Perhaps a compromise is for the authors to acknowledge in the manuscript that they did not analyse all the available brown algal genomes at the ordinal level (I listed these in my last review). This, at least, would make this aspect of the manuscript transparent to the reader.

Response: Thanks for this valuable comment. We added the deficiencies in our research in the conclusion section of the manuscript as follows：

“However, our study did not integrate all brown algae orders, and additional organellar genomes at the ordinal level are needed for further study of their organellar genome evolution.”

General comments: I also took issue with the broader implications and interpretations of this work in my previous review. I think this is because the manuscript is very descriptive. I wonder how useful or relevant it is to present information such as total gene length, space content, or overlap sizes, especially when these numbers are so highly conserved across brown seaweeds. I’m just not clear on how or why these details are relevant. But the authors are entitled to present this, and if the editor is fine with a descriptive study of this nature, then I respect that decision.

Response: Thank you, we believe that the description of the organelle genome is an important part of the analysis of the first published organelle genome, and lays the foundation for the subsequent analysis of the structural variation of the organelle genome.

General comments:While I commend the authors on making corrections regarding gene losses, I must insist (again) that they upload their short read data to the short read archive, in addition to their submission of the fully assembled genomes to genbank (which they have already done). The assembled genomes are an endpoint in your analysis, not the data you generated, the latter of which must be available for re-analysis. The raw data (the short reads you used to map and assemble the organellar genomes) are fundamental to the repeatability of this work (at least, insofar as it relates to reproducing the organellar genomes of S. siliquosa). This must be done before I can endorse for publication.

Response: Thank，the raw reads of Silvetia siliquosa organelle genomes have been deposited in the NCBI Sequence Read Archive under the BioProject number PRJNA824893, Sequence Read Archive accession numbers of mitochondrial and chloroplast genomes are SAMN27488512 and SAMN27488513, respectively.

---

## [Editor Report · Decision Letter 3]

25 May 2022

The organellar genomes of Silvetia siliquosa (Fucales, Phaeophyceae) and comparative analyses of the brown algae

PONE-D-21-32930R3

Dear Duan

We’re pleased to inform you that your manuscript has been judged scientifically suitable for publication and will be formally accepted for publication once it meets all outstanding technical requirements.

Kind regards,

Genlou Sun

Academic Editor

PLOS ONE
---

## [Editor Report · Acceptance letter]

6 Jun 2022

PONE-D-21-32930R3 

The organellar genomes of *Silvetia siliquosa* (Fucales, Phaeophyceae) and comparative analyses of the brown algae 

Dear Dr. Duan:

I'm pleased to inform you that your manuscript has been deemed suitable for publication in PLOS ONE. Congratulations! Your manuscript is now with our production department. 

Kind regards, 

on behalf of

Dr. Genlou Sun 

Academic Editor

PLOS ONE